# Performance of Multi-Band MDE-Based Virtual Sensing for Estimating Lifetime Fatigue Damage Equivalent Loads for the IEA 15 MW Reference Wind Turbine

Mads Greve Pedersen<sup>1,2</sup>, Jennifer Marie Rinker<sup>3</sup>, Isaac Farrereas Alcover<sup>2</sup>, and Jan Høgsberg<sup>1</sup> <sup>1</sup>Department of Civil and of Mechanical Engineering, Technical University of Denmark, 2800 Kongens Lyngby, Denmark <sup>2</sup>COWI A/S, 2800 Kongens Lyngby, Denmark <sup>3</sup>Department of Wind and Energy Systems, Technical University of Denmark, 2800 Kongens Lyngby, Denmark **Correspondence:** Mads Greve Pedersen (madg@dtu.dk)

**Abstract.** Growing Offshore Wind Turbines (OWTs) are increasingly vulnerable to fatigue damage, motivating stress monitoring at critical, often inaccessible locations, for asset integrity management and life-extension. Virtual sensing methodologies, such as multi-band Modal Decomposition and Expansion (MDE), offer a solution by extrapolating measurements from sensors at accessible locations. However, existing MDE studies often model the Rotor-Nacelle-Assembly (RNA) as a lumped mass

inertia, thereby ignoring blade flexibility and rotor operation. This leads to errors in estimated strains or stresses, particularly close to the tower top, where blade vibrations significantly influence the structural response. Moreover, neglecting blade flexibility can also lead to inaccurate tower mode shapes, causing errors not limited to the tower top.

The present paper investigates the errors of multi-band MDE estimates resulting from modelling the RNA as a lumped inertia. To this end, a dataset of HAWC2 simulations covering the Fatigue Limit State (FLS) design life of the IEA Wind

- 15-Megawatt Offshore Reference Wind Turbine with a monopile foundation (IEA 15-MW RWT) is considered. Utilizing this dataset, multi-band MDE is used to estimate section moments along the entire supporting structure of the IEA 15-MW RWT. These estimates are compared against the true response extracted from the dataset in terms of Damage Equivalent Loads (DELs) and Damage Equivalent Stresses (DESs) combined for the individual Design Load Cases (DLCs). Additionally, the error of the MDE estimates is assessed for individual 10-minute time series from the same dataset. Based on the combined
- DELs and DESs, it is concluded that the MDE used in the present work performs well for long-term estimates, except in the area around the tower top, where blade vibrations and 3P effects significantly impact the quality of the estimates. It is shown that the MDE errors for the individual 10-minute time series are generally in the range of  $\pm 5\%$ . However, the error is as high as 180% in the tower top, where the impact from the lumped inertia RNA model is large. Finally, the error of the MDE estimates exhibits wind speed dependency. This underlines the inherent limitation in the MDE, which assumes a linear and
- time-invariant response and thus cannot capture the temporal variability of the dynamic model due to changing operational and environmental conditions. In conclusion, multi-band MDE provides accurate estimates of section moments across most of the IEA 15-MW RWT supporting structure, though without capturing the effects of operational and environmental variability. Furthermore, improvements are necessary to effectively capture the effects of blade flexibility, particularly near the tower top.

# 1 Introduction

- During recent decades, wind turbines have been consistently growing in size, and modern Offshore Wind Turbines (OWTs) planned for deployment, such as the Vestas V236-15MW, now have a power production of up to 15 MW and rotor diameters approaching 240 m. The growth in wind turbine size results in highly flexible supporting structures (tower, transition piece, and foundation), with the lowest natural frequencies approaching the quasi-static frequency domain. This makes them susceptible to dynamic excitation from turbulence and wave loads, resulting in designs that are increasingly vulnerable to fatigue damage
- (Zou et al., 2023). At the same time, the most recent decades have experienced the emergence of Structural Health Monitoring (SHM), where data from sensors installed in a given structure is applied to inform operation and maintenance (O&M) strategies, in asset integrity assessments, and lately also for the assessment of potential life-extension through monitoring of strain histories at fatigue critical locations. However, for offshore structures, these critical locations are often located sub-soil or sub-sea, where strain sensors cannot be installed or maintained post-erection. Furthermore, pre-installed sensors are likely to be damaged
- during erection, while any undamaged strain sensors tend to fail after a few years (Toftekær et al., 2023). To overcome these challenges, virtual sensing has gained traction in SHM of OWTs, where structural responses (stresses or strains) are estimated by so-called virtual sensors, in which physical (above-sea) sensor signals are extrapolated to critical locations by a digital process model.

According to Zou et al. (2023), virtual sensing process models can be separated into two main categories. The deterministic
approach uses model-based extrapolation, from which strain responses are estimated based on measurements from e.g. accelerometers, inclinometers, strain gauges, or 3D point tracking (Baqersad et al., 2015). The alternative probabilistic approach applies state-estimation from Kalman filters (Maes et al., 2016), augmented Kalman filters (Vettori et al., 2023), dual Kalman filters (Eftekhar Azam et al., 2015), or, more recently, from a generic latent force model (Bilbao et al., 2022; Zou et al., 2023). Lately, the use of neural networks has also entered the field of virtual sensing, e.g. when physics-guided learning from SCADA
data and 10-minute acceleration statistics are used to estimate damage equivalent moments (de N Santos et al., 2023).

The present work applies the predominant deterministic model-based expansion method: Modal Decomposition and Expansion (MDE). The concept of virtual sensing by MDE was initially introduced for dynamic strain estimation in OWTs in the pioneering work by Iliopoulos et al. (2014, 2016), and subsequently extended in Iliopoulos et al. (2017) to multi-band MDE, where strain histories are estimated individually in separate frequency bands (quasi-static, low-frequency and high-frequency)

- based on measurements from strain gauges (for the quasi-static band) and accelerometers (for low- and high-frequency bands) using mode shapes and static deflection shapes from a Finite Element (FE) beam model with a lumped Rotor-Nacelle-Assembly (RNA) inertia. This approach has been further developed by Noppe et al. (2016), using a SCADA-driven thrust load model for quasi-static band estimation, and by Henkel et al. (2021) for estimating and validating sub-soil fatigue stresses by dual-band MDE with experimental mode shapes and Operational Deflection Shapes (ODSs).
- The use of experimental ODSs and mode shapes is also applied for strain estimation using a synthetic response of the National Renewable Energy Laboratory (NREL) 5 MW Reference Wind Turbine with an OC4 jacket substructure in Henkel et al. (2020), indicating less good performance for strains in the braces due to the occurrence of local brace modes and extrapolation



of the wave loading. Augustyn et al. (2021) attempts to improve the accuracy for jacket structures by including sensors in a few submerged braces and applying the wave load generated Ritz vectors from Skafte et al. (2017) and local brace modes in MDE.

Recently, Toftekær et al. (2023) have investigated the use of rotations obtained from filtered acceleration measurements in combination with Ritz vectors to estimate quasi-static stresses at the mud line of an 8.4 MW offshore wind turbine, and thereby quantifying the accuracy of the estimated stress range histories for different modal expansion configurations. Subsequently, Fallais et al. (2024) have investigated the accuracy of a single-model MDE configuration for estimating damage equivalent stresses in the lower part of an OWT supporting structure, concluding that varying operational conditions across 2000 10-

minute time series only have a minor impact on the estimate precision.

Studies performing strain/stress estimates for monopile-supported OWTs, using MDE with mode shapes and Ritz vectors from an FE model (Iliopoulos et al., 2017; Noppe et al., 2016; Toftekær et al., 2023; Fallais et al., 2024), commonly consider the RNA as a lumped inertia. Consequently, the tower mode shapes that include blade motions are estimated inaccurately, and

- the influence of blade flexibility and rotor operation (e.g., blade vibrations, 3P effects, and operational variability) on the tower vibrations are not accounted for in the MDE. Given the inherent coupling between the tower and the blades, this simplification can introduce errors in the strains or stresses estimated in the supporting structure. Furthermore, the MDE performance is usually evaluated in the lower part of the supporting structure, where the influence from errors in the RNA model is less pronounced, giving an erroneous impression of their importance. Finally, these studies do not include wave loading separately
- in the MDE, thus assuming that wave loads are either insignificant or that the associated dynamic mode shapes can well capture their effects. However, these simplifications will lead to errors in the estimated strains and stresses in areas of the OWT supporting structure exposed to substantial wave loading.

The present paper addresses the errors associated with representing the rotor by a lumped RNA inertia and its influence on the MDE prediction of Damage Equivalent Loads (DELs) and Stresses (DESs) in modern scale offshore wind turbines.

Furthermore, it investigates how wave loads can be explicitly included in the Ritz vectors for quasi-static and low-frequency estimation. For that precise purpose, uncertainties from soil modelling, variations in the OWT's as-built conditions, and measurement noise from sensors have been eliminated by considering the synthetic response data in Pedersen et al. (2025), which is an open access dataset (available for download at https://doi.org/10.11583/DTU.24460090) containing response simulations covering the Fatigue Limit State (FLS) design life of the IEA Wind 15-Megawatt Offshore Reference Wind Turbine with a monopile foundation (IEA 15-MW RWT) version 1.1.6 (Gaertner et al., 2020a).

The structure of the paper is as follows. Section 2 presents the data from Pedersen et al. (2025), the assessment of the performance of the IEA 15-MW RWT, and a relative lifetime damage calculation made for the individual design load cases included in Pedersen et al. (2025). Section 3 explains the multi-band MDE methodology used in the present work and the Finite Element (FE) model used to extract mode shapes and Ritz vectors for the MDE. In Section 4 the MDE is used for the

estimation of Damage Equivalent Loads (DELs) and Stresses (DESs) and the MDE errors are quantified and discussed, with the final Section 5 providing conclusions and perspective for future work.

# 2 Data

The present work is based on synthetic wind turbine response data from the online open-access dataset "*IEA-15MW-RWT-Monopile HAWC2 Response Database*" (Pedersen et al., 2025), which is available for download at https://doi.org/10.115
83/DTU.24460090 along with the relevant documentation, model- and input files, and scripts for reading and sorting data. The dataset comprises 4902 HAWC2 output files covering the Fatigue Limit State (FLS) design life of the IEA 15-MW RWT version 1.1.6, which is described in Gaertner et al. (2020a). The metocean data used for the simulations performed by Pedersen et al. (2025) is based on the metocean assessment performed for Energinet Eltransmission A/S in DHI (2023a), DHI (2023b), and DHI (2023c). The individual HAWC2 output files contain time series data from 898 sensors hereunder environmental- and operational data (e.g. hub wind speed, wave height, rotor speed, blade pitch angles, torque, thrust, and power production) and structural response data in terms of displacements, rotations, accelerations, forces, and moments in the individual structural

members.


In the following sections, the IEA 15-MW RWT and the Design Load Cases (DLCs) considered in Pedersen et al. (2025) are described briefly, before the assessment of the IEA 15-MW RWT performance is conducted. Finally, the relative lifetime damage from the individual DLCs is calculated for the IEA 15-MW RWT, based on Damage Equivalent Loads (DELs).

### 2.1 IEA Wind 15-Megawatt Offshore Reference Wind Turbine

The IEA 15-MW RWT is a monopile-founded offshore wind turbine with a rated power of 15 MW and a cut-in, rated, and cut-out wind speed of  $V_{in} = 3 \text{ m/s}$ ,  $V_r = 10.69 \text{ m/s}$ , and  $V_{out} = 25 \text{ m/s}$ , respectively. The supporting structure consists of a 75 m monopile with an embedment depth of 45 m, a 15 m transition piece, and a 129.4 m tower, see Figure 1. The design of the supporting structure has been derived from the Ultimate Limit State (ULS) and modal analysis following a soft-stiff approach

supporting structure has been derived from the Ultimate Limit State (ULS) and modal analysis following a soft-stiff approach (Gaertner et al., 2020a), thus locating the natural frequency of approximately 0.17 Hz for the first order tower bending modes between the 1P and 3P rotor frequencies. The design of the IEA 15-MW RWT is available from the Github repository in Gaertner et al. (2023).

### 2.2 Modelling

- As previously stated, the database in Pedersen et al. (2025) comprises synthetic wind turbine response data obtained by HAWC2 simulations, whereby it inherits the limitations and assumptions associated with HAWC2. HAWC2 calculates the aerodynamic loads based on Blade Element Momentum (BEM) theory. The implementation of BEM theory in HAWC2 has been extended to account e.g. for dynamic inflow, dynamic stall, and the rotor's yaw and tilt (Larsen and Hansen, 2021). In the present work, the turbulent wind field is modelled using the Mann Turbulence generator which is directly linked with HAWC2. The tower
- shadow effect is accounted for using a potential flow model, and the wind shear is implemented using the standard power law expression

$$V(z) = V(z_r) \left(\frac{z}{z_r}\right)^{\alpha} \tag{1}$$

**Figure 1.** Overview of the IEA 15-MW RWT (data from Gaertner et al. (2020a). The RWT has a hub height of 150 m above the Mean Sea Level (MSL) and a rotor radius of 120 m. The water depth at the chosen site is 30 m. The supporting structure of the RWT consists of a 75 m monopile with an embedment depth of 45 m, a 15 m transition piece, and 129.4 m tower.

where V(z) is the wind speed across the elevation z above the Mean Sea Level (MSL),  $z_r$  is the reference elevation at which the wind speed  $V(z_r)$  is known (in this case at hub-height), while  $\alpha = 0.08$  from the metocean assessment in DHI (2023a).

The structural modelling in HAWC2 is based on a multi-body formulation, where each body is an assembly of Timoschenko beam elements. Thus, the formulation for the structural members accounts for large deflections and rotations, geometrical non-linearities, and shear deformations (Larsen and Hansen, 2021). The soil model implemented in the model for simulations performed by Pedersen et al. (2025) utilize the lateral linear soil springs presented in Table 1. In HAWC2, the hydrodynamic forces acting on the monopile are calculated using Morison's formula. The present work ignores the current when calculating

hydrodynamic forces, and the water kinematics are calculated based on the irregular Pierson–Moskowitz wave spectrum, utilising the significant wind speed-dependent wave height and the wave period from the metocean assessment in DHI (2023c).