# Peer review of "Performance of Multi-Band MDE-Based Virtual Sensing for Estimating Lifetime Fatigue Damage Equivalent Loads for the IEA 15 MW Reference Wind Turbine"

_Wind Energy Science, 2025_

## Referee Comment (RC2)

[revised manuscript text omitted]

**2.3 Load Cases**

The Design Load Cases (DLCs) for the Fatigue Limit State (FLS) of bottom-fixed OWTs are described in IEC 61400-3-1:2019 (IEC, 2019b). In Pedersen et al. (2025), the implementation of the DLCs follows Natarajan et al. (2016), with the input values used for the HAWC2 simulations presented in Table 2. The number of simulations in Table 2 is a result of the operational and environmental variability needed to capture the individual load cases, e.g. DLC 1.2 considers 11 different *wind speeds* at three different *yaw errors*, *wind-wave misalignments*, and *Mean Water Levels (MWL)*. Finally, six seeds are used to secure numerical robustness for the simulation of both turbulence and irregular waves. In total, this gives $11 \times 3 \times 3 \times 3 \times 6 = 1782$ simulations for DLC 1.2. According to DHI (2023b), the tidal effects at the chosen site are weak and thus only the simulations where the Mean Water Level (MWL) is equal to the Mean Sea Level (MSL) are considered, thereby discarding simulations where MWL is at either Lowest (LAT) or Higest (HAT) Astronomical Tide in the analysis conducted for the present paper.

To evaluate the lifetime damage contribution from the individual HAWC2 simulations, their representative durations are calculated based on the joint probability of the DLC occurrence and the environmental parameters: Wind speed, yaw error, and wind-wave misalignment. An overview of the input for the duration of the individual simulations is presented in Table 3. The duration of the individual DLCs is based on the recommendations in Section 7 of IEC (2019b). The application of these recommendations in the present work is presented below.

- DLC 1.2: It is expected that the wind turbine will be available for operation at normal conditions for 90 % of its 20-year lifetime. In the present work, this is interpreted as DLC 1.2 occurring 90 % of the time the wind speed falls within the cut-in and cut-out wind speed ($V_{in} = 3$ m/s and $V_{out} = 25$ m/s).

[Figure]

[Figure]

**Table 2.** Overview of DLCs from IEC (2019b) considered in Pedersen et al. (2025).

| DLC | Description | Environmental parameters | | | No. Simulations |
|---|---|---|---|---|---|
| 1.2 | Power production in normal conditions | Wind speed | [4:2:24] | [ m/s] | 1782 |
| | | Yaw error | -10, 0, 10 | [deg] | |
| | | wind-wave misalignment | -22.5, 0, 22.5 | [deg] | |
| | | Sea level | LAT, MSL, HAT | [m] | |
| 2.4 | Power production with large yaw errors in normal conditions | Wind speed | [4:2:24] | [ m/s] | 132 |
| | | Yaw error | -20, 20 | [deg] | |
| | | wind-wave misalignment | 0 | [deg] | |
| | | Sea level | MSL | [m] | |
| 3.1 | Start-up in normal conditions | Wind speed | 3, 10.69, 25 | [ m/s] | 18 |
| | | Yaw error | 0 | [deg] | |
| | | wind-wave misalignment | 0 | [deg] | |
| | | Sea level | MSL | [m] | |
| 4.1 | Shut-down in normal conditions | Wind speed | 3, 10.69, 25 | [ m/s] | 18 |
| | | Yaw error | 0 | [deg] | |
| | | wind-wave misalignment | 0 | [deg] | |
| | | Sea level | MSL | [m] | |
| 6.4 | Parked turbine with idle rotor in normal conditions | Wind speed | [4:2:34] | [ m/s] | 576 |
| | | Yaw error | -8, 8 | [deg] | |
| | | wind-wave misalignment | 0 | [deg] | |
| | | Sea level | LAT, MSL, HAT | [m] | |
| 7.2 | Fault - locked rotor at azimuth angle $0°, 30°, 60°,$ and $90°$ in normal conditions | Wind speed | [4:2:24] | [ m/s] | 2376* |
| | | Yaw error | -10, 0, 10 | [deg] | |
| | | wind-wave misalignment | 0 | [deg] | |
| | | Sea level | LAT, MSL, HAT | [m] | |

*208 simulations of the simulations for DLC 7.2 failed to converge and are disregarded in the further work.

150     – DLC 2.4: For operation during the occurrence of fault or loss to the electrical network, IEC (2019b) suggests that the duration may be applied as follows: 10 shut-downs per year for overspeed event, 24 hours per year of operation for events with yaw error, 24 hours per year of operation for events with pitch error, and 20 times per year with loss of electrical network connection. In Pedersen et al. (2025) only the fault "*operation for events with yaw error*" is modelled.

[Figure]

**Table 3.** Input for joint probability used for calculating the expected life-time duration for the individual time series available in Pedersen et al. (2025).

| DLC | Exposure | Wind speed | Yaw error | Wind-wave misalignment |
|---|---|---|---|---|
| 1.2 | 90 % | $p(V)$ **for** $V \in [\ 3, 25]\,$m/s | 1/4 , 1/2 , 1/4 | 1/3, 1/3, 1/3 |
| 2.4 | 0.57 % | $p(V)$ **for** $V \in [\ 3, 25]\,$m/s | 1/2 , 1/2 | 1 |
| 3.1 | 0.35 % | 1000/1100 , 50/1100 , 50/1100 | 1 | 1 |
| 4.1 | 0.35 % | 1000/1100 , 50/1100 , 50/1100 | 1 | 1 |
| 6.4 | | $p(V)$ **for** $V \in [25, 35]\,$m/s | 1/4 , 1/2 , 1/4 | 1 |
| 7.2 | 8.7 % | $p(V)$ **for** $V \in [\ 3, 25]\,$m/s | 1/4 , 1/2 , 1/4 | 1 |

To account for the damage occurring during the remaining fault conditions specified for DLC 2.4, the duration is adjusted to 50 hours per year of operation (0.57 % of the time the wind speed falls within the $V_{in}$ and $V_{out}$ ) in the present work.

– DLC 3.1 and 4.1: IEC (2019b) states that start-up/shut-down in normal conditions (DLC 3.1/4.1) can be expected to occur 1100 times annually: 1000 times at the cut-in wind speed, 50 times at the rated wind speed and 50 times at the cut-out wind speed (0.35 % of the total life for each of DLCs 3.1 and 4.1).

– DLC 6.4: In the present analysis, DLC 6.4 is considered to occur only when the wind speed at the hub exceeds the cut-out wind speed $V_{out} = 25$ m/s. As this DLC is the only one expected to occur for wind speeds above $V_{out}$, the duration of DLC 6.4 is assumed to be the total duration the hub wind speed exceeds the cut-out wind speed.

– DLC 7.2: As IEC (2019b) does not specify a duration for DLC 7.2, this work defines its duration as the time not accounted for by previous DLCs within the operational wind speed range from $V_{in}$ to $V_{out}$, which is 8.7 %.

The wind speed's probability density is assumed to follow the Weibull distribution

$$p(V) = \frac{k}{A} \left(\frac{V}{A}\right)^{k-1} \exp\left(-\left(\frac{V}{A}\right)^{k}\right) \tag{2}$$

with the omnidirectional Weibull parameters $k = 2.35$ and $A = 9.91$ m/s given in DHI (2023b) for a mean wind speed $\bar{V}_{10} = 8.79$ m/s at 10 m above MSL. These values are corrected for the hub height using a wind shear for the Normal Wind Profile (NWP) presented in (1). According to IEC (2019b), only part of the wind speed spectrum is considered, namely $V_{hub} \in [V_{in}, V_{out}]$ for DLC 1.2, 2.4, 3.1, 4.1, and 7.2 and $V_{hub} \in [V_{out}, 0.7 V_{ref}]$ for DLC 6.4. As such, it is assumed that there is no contribution to the fatigue life consumption for $V_{hub} \notin [V_{in}, 0.7 V_{ref}]$, where $V_{ref} = 50$ m/s is the reference wind speed for wind turbine class 1 (IEC, 2019a).

Although the DLCs described above do not exhaustively represent the scenarios occurring during the actual lifetime of an OWT, they provide an overview of the fatigue-life impact from the most common and governing operating scenarios.

[Figure]

[Figure]

**2.4 Performance of the IEA 15-MW RWT**

When performing Modal Decomposition and Expansion (MDE) modal truncation is needed due to a limited number of sensors. Furthermore, a finite number of Ritz vectors can be included to assess the quasi-static part of the response. Hence, it is important to have an overview of the different governing loads to be accounted for in the response estimates. This section gives an example of how diverse operational and environmental conditions can impact the Damage Equivalent Loads (DELs) of the IEA 15-MW RWT, and hence contribute differently to lifetime damage. Specifically, statistical values of relevant operational parameters and the tower base Fore-Aft (FA) and Side-Side (SS) section moments are considered during normal power production (DLC 1.2).

In Figure 2, the statistics (minimum, mean, maximum) of the operational parameters (rotor speed, electrical power, generator torque, thrust, and pitch angle) and the wave amplitude are presented, while Figure 3 shows the associated statistics of the tower base FA and SS section moments and the 1 Hz Damage Equivalent Loads (DELs) for the individual HAWC2 time series (evaluated by (6)) for DLC 1.2. The operational parameters in Figure 2 are compared with steady-wind rotor performance values from Gaertner et al. (2023), generated by the *Wind-plant Integrated System Design and Engineering Model* (WISDEM), which uses the aeroelastic code OpenFAST.

Figure 2(a-e) shows that the mean values generally coincide well with the WISDEM output, and Figure 2(f) verifies that the minimum- and maximum wave amplitudes follow the development of the input significant wave height. The greatest discrepancies are observed for the thrust in Figure 2(d) and the pitch angle in Figure 2(e). The discrepancies in the thrust and pitch angle are due to: steady versus turbulent operation and the ElastoDyn beam model used in the WISDEM calculation (Gaertner et al., 2020b) not including a torsional degree of freedom (Rinker et al., 2020). The generally good match between the models indicates that the HAWC2 model may be used for further analysis.

The statistical values for the tower base FA moment presented in Figure 3(a) follow the thrust curve from Figure 2(d) as expected. The DELs associated with the tower base FA moment presented in Figure 3(c) generally increase with both the wind speed and turbulence. However, they plateau at wind speeds from approximately $12 - 16$ m/s, in which range the blades start to pitch (see Figure 2(e)). This illustrates that the DELs in the FA direction at the tower base are primarily governed by quasi-static wind loading, while operational parameters (e.g., the pitch angle) also affect the damage. Similarly to the statistical values of the tower base FA moment, the mean values of the tower base SS moment presented in Figure 3(b) follow the generator torque curve in Figure 2(c). The minimum and maximum values of the tower base SS moment are symmetric around the mean value with increasing amplitudes for increasing wind speeds. The associated DELs in Figure 3(d) also increase with the wind speed and turbulence. Furthermore, Figure 3(d) shows that the variance of the DELs increases with the wind speed up to the rated wind speed, from where it is rather significant.

To assess the cause of the high variance, the time histories of the tower base SS moment, wind speed (in the SS direction), and wave height associated with the minimum and maximum DELs for the wind speed of $14$ m/s are presented in Figure 4. Considering the moment time series in Figure 4(a) and the related PSD in Figure 4(b), it is concluded that DELs are mainly driven by the first tower SS mode. There is not a significant difference in the frequency content of the wind around the natural

[Figure]

**Figure 2.** Statistical values (minimum, mean, maximum) for selected operational parameters (a) rotor speed, (b) electrical power, (c) generator torque, (d) thrust load, (e) pitch angle, and (f) wave amplitude depicted across the wind speed at the hub, calculated for the HAWC2 time series covering DLC 1.2 for the MWL equal to MSL.

frequencies of the first order tower bending modes. However, the mean wind speed in the SS direction is significantly higher for the maximum DEL than for the minimum DEL, due to the $-10°$ yaw error. Furthermore, the waves have an angle-of-attack of $-32.5°$ for the maximum DEL, whereas it is $0°$ for the minimum DEL. Thus, the variation in DEL magnitude is caused by the excitation of the first tower SS mode occurring for the maximum DEL, while not for the minimum DEL, likely due to

210

[Figure]

**Figure 3.** Statistical values (minimum, mean, maximum) of the tower base moment calculated in (a) the FA direction and (b) the SS direction, and DELs calculated in (c) the FA direction and (d) the SS direction, all based on the HAWC2 time series covering DLC 1.2 for the MWL equal to MSL.

the difference of the excitation forces resulting from the varying angle-of-attack of the wind and waves between the two time series.

In conclusion, the present section underlines that the DELs calculated for the IEA 15-MW RWT are indeed influenced by environmental parameters such as turbulence, which govern the quasi-static response, and wave direction. Furthermore, operational parameters such as pitch angles and yaw errors can, in some cases, contribute to the excitation of the dynamic modes, which significantly impacts the DELs. Thus, the MDE configuration presented in Section 4.1, is required to accurately capture both quasi-static and dynamic responses for varying operational and environmental conditions.

[Figure]

[Figure]

**Figure 4.** Time series data for maximum and minimum DELs from Figure 3(d) at 14 m/s hub wind speed: (a) time history and (b) PSD of tower base moment in the SS direction, (c) time history and (d) PSD of hub wind speed in the SS direction, and (e) time history and (f) PSD of wave amplitude (water surface elevation).

[Figure]

[Figure]

**2.5 Relative lifetime damage results**

220 The present section investigates the lifetime damage of the IEA 15-MW RWT caused by the individual design load cases presented in Section 2.3, thereby giving an overview of which operating scenarios are significant for the fatigue damage in the supporting structure.

According to Veldkamp (2006), the relative lifetime damage caused in a given structure by a load case $i$ is given as

$$d_{i,rel} = \frac{n_i \left(\Delta P_{eq,i}\right)^m}{n_T \left(\Delta P_{eq}\right)^m} \tag{3}$$

225 where $\Delta P_{eq,i}$ represents the 1 Hz DEL ranges for the individual load case $i$, $m$ is the Wöhler coefficient, $n_i$ is the number of 1 Hz cycles for load case $i$, $n_T$ is the total number of 1 Hz cycles in the structure's lifetime, and $\Delta P_{eq}$ is the lifetime DEL range.

In the present analysis, a similar approach to that of Veldkamp (2006) in (3) is used for the evaluation of the relative lifetime damage for individual DLCs. By adding the 1 Hz DELs from the HAWC2 simulations contained in a DLC, the relative damage 230 of the individual DLCs is calculated as

$$d_{DLC,rel} = \frac{\sum_{s \in \text{DLC}} n_s \left(\Delta P_{eq,s}\right)^m}{n_T \left(\Delta P_{eq}\right)^m} \tag{4}$$

where

$$n_s = p(\text{DLC}, V, \theta_{yaw}, \theta_{wwm}) \frac{n_T}{n_{seed}} \tag{5}$$

is the number of 1 Hz cycles during the lifetime of the IEA 15-MW RWT, $p(\,)$ is the joint probability of the input parameters for 235 the operational and environmental conditions (DLC, wind speed ($V$), yaw error ($\theta_{yaw}$), and wind-wave misalignment ($\theta_{wwm}$)) used for the simulation $s$, and $n_{seed}$ is the number of simulations that share these operational and environmental conditions. Note that the number of summations in (4) refers to the number of (converged) simulations in Table 2 for a given DLC at MWL equal to MSL. Finally, in (4) the 1 Hz DEL range for the individual HAWC2 simulations is evaluated as

$$\Delta P_{eq,s} = \left( \frac{\sum n_j \Delta P_j^m}{n_{eq}} \right)^{\frac{1}{m}} \tag{6}$$

240 where $n_{eq}$ is the number of 1 Hz cycles in the time series $s$, while $\Delta P_j$ and $n_j$ are the binned load ranges and corresponding number of load cycles identified from the individual time series using the Rainflow counting method from ASTM E1049-85 (2017). In the present work, a single slope S-N curve with a Wöhler coefficient of $m = 5$ is used for the supporting structure. This is based on $m_1$ of the S-N curves for welded and non-welded circular hollow sections from Chapter 8 in DSF/FprEN 1993-1-9 (2024), which is not representative of the damage at all locations in the supporting structures but still considered 245 sufficiently accurate for the assessment of the impact of the individual DLCs.

The relative damage for the individual DLCs $d_{DLC,rel}$, is presented in Figure 5 for the FA and SS direction of the IEA 15-MW RWT supporting structure. From these relative damage plots in Figure 5, it is observed that there is a big resemblance in the distribution of damage across the height of the IEA 15-MW RWT for DLC 1.2 and 2.4, which is expected as these

[Figure]

**Figure 5.** Relative damage for the individual DLCs calculated across the height of the IEA 15-MW RWT supporting structure as presented in (4) for the FA (a) and SS (b) direction.

load cases are both for operation in normal conditions. A similar expected resemblance is found for DLC 3.1 and 4.1, as these load cases represent start-up and shut-down, respectively. Figure 5(a) shows that approximately 99 % of the damage in the FA direction is caused by DLC 1.2 (Power production in normal conditions), DLC 6.4 (Parked - idle rotor in normal conditions), and 7.2 (Fault - locked rotor in normal conditions). In the SS direction, shown in Figure 5(b), the damage from DLC 6.4 falls below 1 %, so only DLC 1.2 and 7.2 are considered significant for the damage in the SS direction. As presented in Table 3, DLC 1.2 is significantly more frequent than DLC 6.4 and 7.2, and the significant damage contribution of this DLC is associated

with the large duration, whereas for DLC 6.4 and 7.2, the substantial damage contribution is associated with the large DELs (see Figure 12).

The relative damage in the FA direction in Figure 5(a) is dominated by DLC 1.2 at the tower top ($\approx 100 - 144\,\mathrm{m}$ above the MSL). This is due to 3P effects (tower shadow, wind shear, and turbulence), which are significant contributors to damage in the tower top, as the varying forces on the blades and uneven loading on the rotor result in a significant moment at the hub. In the remainder of the free standing supporting structure ($\approx -30 - 100\,\mathrm{m}$), the relative damage in the FA direction is dominated by DLC 7.2. In this area, the section moments are to a higher degree governed by the global bending of the supporting structure caused by the thrust loading (for DLC 1.2) and especially the first tower FA mode (for DLC 6.4 and 7.2). The tower bending modes in the FA direction for DLC 1.2 are subject to significant aerodynamic damping arising from the operating rotor, thus explaining the smaller contribution to the relative damage from this DLC, and the larger contribution from DLCs 6.4 and 7.2, where the rotor is not operating and the aerodynamic damping is effectively negligible. Below MSL the relative damage contribution from DLC 1.2 and 7.2 approaches each other and balances out at the mud line. This is likely due to the influence of wave loads, which increase with the water depth and are less affected by the aerodynamic damping present for DLC 1.2.

The relative damage in the SS direction in Figure 5(b), is dominated by DLC 7.2 at the tower top ($\approx 120 - 144\,\mathrm{m}$ above MSL), while DLC 1.2 dominates the damage below this area. Unlike the FA response, the SS response for DLC 1.2 is not significantly affected by aerodynamic damping, the 3P effects, or the thrust load variations. Consequently, the damage in both DLC 1.2 and 7.2 is primarily driven by ambient excitation at the turbine's resonant frequencies. However, the locked rotor condition in DLC 7.2 particularly influences damage at the tower top. Because the rotor is fixed in rotation and the blades are pitched $90°$, the blades are more susceptible to turbulence-induced excitation, which creates a moment at the blade root. This, in turn, excites the second tower SS mode, and possibly different rotor modes, resulting in DLC 7.2's dominant contribution to damage in the upper part of the supporting structure. In the lower part, the damage patterns are more governed by the first order tower bending modes, which are similar for DLC 1.2 and 7.2. However, the significantly longer duration of DLC 1.2 ($90\,\%$ of the turbine's lifetime) results in it being dominant below $120\,\mathrm{m}$. This effect is visible in Figure 5(b), where the distribution of relative damage from DLCs 1.2 and 7.2 remains rather constant in the supporting structure below $100\,\mathrm{m}$, with $d_{DLC,rel}$ for the two DLCs varying between $69 - 80\,\%$ and $19 - 29\,\%$, respectively.

In conclusion, the damage in the supporting structure of the IEA 15-MW RWT is governed by both normal operation conditions and conditions where the rotor is idling or locked, whereas start-up and shut-down of the wind turbine and operation with yaw error are less critical. The damage contribution across the elevation of the supporting structure arises from different local and global effects caused by different environmental and operational scenarios e.g. turbulence, 3P effects, wave loads, and inherent dynamical properties. It should be emphasised that the durations used in this analysis for the DLCs are estimated, and scenarios can occur where the durations are differently distributed between the DLCs. Therefore, it is also relevant to evaluate DELs for individual DLCs, without accounting for their specific durations, when assessing how the different operational scenarios impact lifetime damage, as done in Section 4.2.

[Figure]

**3 Virtual Sensing**

The present section initially explains the basic concepts of multi-band MDE and the methodology applied when moving from
290  nodal displacements to internal force estimates. This is followed by a presentation of the prediction FE model used in the
subsequent estimation of Damage Equivalent Loads (DELs) and Stresses (DESs) in Section 4. Finally, the current section
presents the model output with respect to dynamic mode shapes and quasi-static Ritz vectors.

**3.1 Modal Decomposition and Expansion**

Modal Decomposition and Expansion (MDE) is a well-established process model in virtual sensing (see Section 1). The
295  formulation used in the present work is described in Iliopoulos et al. (2017). MDE assumes that the displacement vector $\mathbf{u}(t)$
of an undamped dynamic system can be decomposed and written as a linear combination of the system's mode shapes and
modal coordinates on the matrix form

$$\mathbf{u}(t) = \mathbf{\Phi}\mathbf{q}(t) \tag{7}$$

The mode shape matrix $\mathbf{\Phi} = [\boldsymbol{\varphi}_1, \boldsymbol{\varphi}_2, \cdots, \boldsymbol{\varphi}_n]$ contains the $n$ mode shapes ($\boldsymbol{\varphi}_j$) included to describe the dynamical system,
300  while the modal coordinate vector $\mathbf{q}(t) = [q_1(t), q_2(t), \cdots, q_n(t)]^T$ collects the corresponding modal coordinates ($q_j$) at each
time instant $t$. The mode shapes of the system $\boldsymbol{\varphi}_j$, can be derived from, e.g., experimental or operational modal analysis, while
in the present work, the vectors $\boldsymbol{\varphi}_j$ are derived from an FE model representing the dynamic system in Section 3.3. Assuming
that the FE model is an accurate representation of the considered dynamic system, it follows that

$$\mathbf{\Phi} = \mathbf{\Phi}_{FE} \tag{8}$$

305  which applies in the remainder of the paper. If the total number of Degrees of Freedom (DOFs) in the FE model is $n_{dof}$, the
modal matrix $\mathbf{\Phi}$ becomes an $n_{dof} \times n$ array. The nodal displacement vector $\mathbf{u}(t)$ in (7) is conveniently partitioned as

$$\mathbf{u}(t) = \begin{bmatrix} \mathbf{u}_m(t) \\ \mathbf{u}_p(t) \end{bmatrix} = \begin{bmatrix} \mathbf{\Phi}_m \\ \mathbf{\Phi}_p \end{bmatrix} \mathbf{q}(t) \tag{9}$$

where the first $n_m$ DOFs in $\mathbf{u}_m(t)$ represent those that are measured by physical sensors, while the remaining $n_p$ DOFs in
$\mathbf{u}_p(t)$ are those that are predicted by the MDE, i.e. the virtual sensors. By direct comparison of (7) and (9), the mode shape
310  matrix is similarly partitioned into

$$\mathbf{\Phi} = \begin{bmatrix} \mathbf{\Phi}_m \\ \mathbf{\Phi}_p \end{bmatrix} \tag{10}$$

in which the $n_m \times n$ array $\mathbf{\Phi}_m$ refers to the mode shape amplitudes associated with the measured DOFs, while correspondingly
the $n_p \times n$ array $\mathbf{\Phi}_p$ accounts for the remaining DOFs that are used for the subsequent prediction procedure. From the above
partitioning in (9) and (10), it is seen that the total number of DOFs in the FE model is $n_{dof} = n_m + n_p$, i.e. the sum of
315  measured and predicted DOFs.

[Figure]

[Figure]

MDE utilises that the displacements in $\mathbf{u}_m(t)$ are available from measurements, while the remaining DOFs in $\mathbf{u}_p(t)$ are predicted simultaneously once the modal matrix in (10) can be obtained from the underlying FE-model with sufficient accuracy. It follows from (9) that the predicted nodal displacements can be expressed by the modal representation

$$\mathbf{u}_p(t) = \boldsymbol{\Phi}_p \mathbf{q}(t) \tag{11}$$

320 The modal coordinates in $\mathbf{q}(t)$, used for the extrapolation in (11), are determined by the corresponding relation

$$\mathbf{u}_m(t) = \boldsymbol{\Phi}_m \mathbf{q}(t) \tag{12}$$

for the measured DOFs in $\mathbf{u}_m(t)$. The inversion of this relation requires that the dynamic displacement field can be represented by at most $n$ modes, where $n$ must be less than or equal to the number of measured DOFs $n_m$. Hereby, the modal coordinates can be determined as

325 $$\mathbf{q}(t) = \boldsymbol{\Phi}_m^\dagger \mathbf{u}_m(t) \tag{13}$$

using the Moore-Penrose pseudo-inverse depicted by the commonly used $(\ )^\dagger$ symbol. The predicted nodal displacements are then obtained by substitution of (13) into (11), which then takes on its final form

$$\mathbf{u}_p(t) = \boldsymbol{\Phi}_p \boldsymbol{\Phi}_m^\dagger \mathbf{u}_m(t) \tag{14}$$

In virtual sensing, one of the objectives is to minimise the number of physical sensors $n_m$ by introducing virtual sensors.
330 Hence, the condition $n \leq n_m$ poses a challenge, as this limits the number of modes $n$ that can be included to describe the dynamic system. Furthermore, for low frequencies, it can be desirable to perform MDE using only a subset of the measurements $\tilde{\mathbf{u}}_m(t)$ to minimise the noise introduced in the estimates, or to introduce Ritz vectors containing static deflection shapes to predict the response $\mathbf{u}_p(t)$ in frequency ranges not dominated by resonant response (see Section 3.3.2). The introduction of multi-band virtual sensing in Iliopoulos et al. (2017) utilises that the nodal displacement vector $\mathbf{u}(t)$ can be divided into separate
335 bands $B_i$ in the frequency domain, which when combined by summation, reattains the original nodal displacement vector

$$\mathbf{u}(t) = \sum_{i=1}^{N} \mathbf{u}_i(t) = \sum_{i=1}^{N} B_i(\mathbf{u}(t)) \tag{15}$$

where $\mathbf{u}_i$ is the nodal displacement vector band-pass filtered in the band $B_i$, and $i = 1, 2 \ldots N$ denotes the individual frequency bands, shown in Figure 6. Similarly, the predicted nodal displacements $\mathbf{u}_p(t)$ can be calculated in individual bands and combined by summation as

340 $$\mathbf{u}_p(t) = \sum_i \mathbf{u}_{i,p}(t) = \sum_i \tilde{\boldsymbol{\Phi}}_{i,p} \tilde{\boldsymbol{\Phi}}_{i,m}^\dagger \tilde{\mathbf{u}}_{i,m}(t) \tag{16}$$

now only including the modes and Ritz vectors $\tilde{\boldsymbol{\Phi}}_i$ and the measurements $\tilde{\mathbf{u}}_m(t)$ that are relevant for the band $B_i$. This representation assumes that the energy content of $\mathbf{u}_p(t)$ is fully captured by the sum of its filtered components in the bands $B_i$.

[Figure]

[Figure]

**Figure 6.** Normalized PSD of moment time series (from DLC 1.2). The frequency spectra of the moments at the yaw bearing, tower base, and mud line are shown in the FA and SS directions. Transparent white/grey bands indicate the frequency ranges used in the MDE (Section 4.1), representing: Band 1 (turbulence), Band 2 (turbulence and wave loads), Band 3 (first tower bending and wave loads), and Band 4 (higher dynamic modes and rotor harmonics).

**3.2 Internal force estimation**

The previous Section 3.1 has explained how modal decomposition and expansion can be used to predict displacement re-
345 sponse at virtual sensor locations. The present section extends the MDE to predict internal forces based on the predicted nodal displacement vector $\mathbf{u}_p(t)$.

The section forces to be predicted by the proposed method are specific for the element of the applied FE representation, e.g. bending moments for the planar beam elements used to describe the dynamics of the present supporting structure. Let the nodal forces be contained in the nodal element vector

$$\quad \mathbf{r}_e(t) = \begin{bmatrix} \mathbf{r}_A(t) \\ \mathbf{r}_B(t) \end{bmatrix}_e = \begin{bmatrix} f_x^A(t), & f_y^A(t), & m^A(t), & f_x^B(t), & f_y^B(t), & m^B(t) \end{bmatrix}_e^T \tag{17}$$

$$= \begin{bmatrix} -N_A(t), & V_A(t), & -M_A(t), & N_B(t), & -V_B(t), & M_B(t) \end{bmatrix}_e^T$$

for a planar (2D) beam element $e$ between two nodes $A$ and $B$, with $f_x$, $f_y$ and $m$ representing the nodal normal force, shear force and moment, respectively. As shown in (17), the corresponding section forces $N$, $V$ and $M$ are derived from the nodal force by appropriate sign changes.

355 For a given element (subscript) $e$, the element nodal force vector in (17) can be determined by the element stiffness matrix $\mathbf{k}_e$. The element stiffness relation can thus be written as

$$\mathbf{r}_e(t) = \mathbf{k}_e \mathbf{T}_e \mathbf{u}_p(t) \tag{18}$$

where $\mathbf{T}_e$ is a $6 \times n_p$ array that both collects and rotates the six DOFs from the global vector $\mathbf{u}_p(t)$ into the local coordinate system for element $e = 1, 2 \ldots N_e$. Elimination of the response in the predicted DOFs $\mathbf{u}_p(t)$ by (14) gives the compact representation

$$\mathbf{r}_e(t) = \mathbf{k}_e \mathbf{T}_e \mathbf{\Phi}_p \mathbf{\Phi}_m^{\dagger} \mathbf{u}_m(t) = \mathbf{D}_e \mathbf{u}_m(t) \tag{19}$$

where

$$\mathbf{D}_e = \mathbf{k}_e \mathbf{T}_e \mathbf{\Phi}_p \mathbf{\Phi}_m^{\dagger} \tag{20}$$

defines the section force matrix that predicts the section forces $\mathbf{r}_e(t)$ from the measured nodal displacements in $\mathbf{u}_m(t)$. For a model with vertical beam elements, as in the present case, the transformation matrix $\mathbf{T}_e$ is an all-zero $6 \times n_p$ matrix, except for $\pm 1$ entries in the $6 \times 6$ block associated with the specific element $e$.

**3.3 Prediction FE model**

The prediction FE model from which the mode shapes and Ritz vectors used in the MDE are obtained is a 3D linear elastic beam model with the Rotor-Nacelle-Assembly (RNA) and transition piece modelled as lumped inertias. The beam model is presented schematically in Figure 7. The geometrical properties and the mass and stiffness input parameters for the prediction FE model are extracted from the HAWC2 model of the IEA 15-MW RWT described in Section 2.1 and presented in Appendix A.

The beam element stiffness is established according to Krenk and Høgsberg (2013), which combines the element stiffness matrix developed from the Timoshenko beam theory $\mathbf{K}_{beam,e}$ with a so-called geometric stiffness term $\mathbf{K}_{g,e}$ expressing the total element stiffness matrix as

$$\mathbf{K}_e = \mathbf{K}_{beam,e} + \mathbf{K}_{g,e} \tag{21}$$

thus accounting for the stiffness contribution adhering from the normal forces causing Euler buckling in bending, although omitting the stiffness terms associated with torsion, i.e., loads causing lateral buckling in static analysis.

The monopile foundation support conditions are modelled using lateral linear elastic soil springs in the embedded part of the monopile. The stiffness of the individual springs $k_{soil,n}$ varies with the embedment depth, as presented in Table 1. The bottom node in the beam model restrains torsion and vertical translation.

The mass contributing to the modal mass of the prediction FE model includes the distributed mass of the tower, transition piece, and monopile presented in Appendix A, the nodal mass of the transition piece $M_{TP}$ located at the top of the transition piece, and the eccentric nodal mass and inertia tensor of the RNA, $M_{RNA}$ and $I_{RNA}$, located at the distances $a_x$, $a_y$, and $a_z$ relative to the top of the tower. The input parameters for the nodal masses for the TP and RNA and the mass moments and mass products of inertia included in the inertia tensor ($I_{xx}, I_{yy}, I_{zz}, I_{xy}, I_{xz}, I_{yz},$) are presented in Table 4. In addition to the mass contributions already presented, an external mass contribution referred to as the hydrodynamic mass $m_{hydro}$ arises when a body moves in a fluid. According to Sumer and Fredsøe (1997), the hydrodynamic mass per unit length of a free circular

[Figure]

[Figure]

[Figure]

**Figure 7.** Schematic presentation of the prediction FE model used for the modal decomposition and expansion, including the height of the members in the supporting structure $h_*$, the element stiffness $\mathbf{K}_{beam,e} + \mathbf{K}_{g,e}$, the nodal masses of the transition piece $M_{TP}$ and Rotor-Nacelle-Assembly (RNA) $M_{RNA}$, the RNA inertia tensor $I_{RNA}$, the soil stiffness in node $n$ $k_{soil,n}$, and the hydrodynamic added mass $m_{hydro}$.

cylinder can be expressed as

$$m_{hydro} = \rho \, C_m \, A \tag{22}$$

if the current is disregarded. Here, the fluid density is $\rho = 1027$ kg/m$^3$, $C_m = 1$ is the hydrodynamic mass coefficient for a cylinder, and $A = \pi r^2$ is the fluid-displaced area for the monopile with radius $r$.

The first three tower bending mode shapes used for the MDE configuration in Section 4.1 have been calculated using the FE model presented above. They are shown in Figure 8 for displacements and bending moments in the FA and SS directions.

[Figure]

[Figure]

**Table 4.** Nodal mass, inertia tensor, and Center of Gravity (CoG) of the IEA 15-MW RWT RNA, calculated based on the individual body properties extracted from HAWC2 and nodal mass of the IEA 15-MW RWT Transition Piece (TP).

| | | |
|---|---|---|
| $M_{RNA}$ | 9.45E+05 | [kg] |
| $a_x$ | -7.12E+00 | |
| $a_y$ | 0 | [m] |
| $a_z$ | 4.58E+00 | |
| $I_{xx}$ | 3.52E+08 | |
| $I_{yy}$ | 1.96E+08 | |
| $I_{zz}$ | 1.97E+08 | |
| $I_{xy}$ | 0 | [kgm$^2$] |
| $I_{xz}$ | -4.04E+07 | |
| $I_{yz}$ | 0 | |
| $M_{TP}$ | 1.00E+05 | [kg] |

[Figure]

**Figure 8.** Mode shapes in terms of displacement and bending extracted from the prediction FE model presented in Figure 7 in the FA and SS direction: (a) the first tower bending modes, (b) the second tower bending modes, and (c) the third tower bending modes.

[Figure]

[Figure]

**3.3.1 Model Validation**

In the present section, the prediction FE model presented in the previous Section 3.3 is validated. The validation is performed simply by comparing the undamped natural frequencies $f_n$ of the prediction FE model to those of the IEA 15-MW RWT extracted using the HAWC2 built-in module *system_eigenanalysis*. The objective of the validation is to ensure that the input parameters for the prediction FE model presented in Figure 7, which are extracted from the HAWC2 model, are interpreted correctly. To ensure that the present validation is as objective as possible, the comparison is performed for a simplified HAWC2 model of the IEA 15-MW RWT, in which particular flexibilities are restrained.

As mentioned previously, the prediction FE model does not include a detailed model of the RNA. Therefore, the influence of an operating rotor, blade flexibility, and shaft torsion is not included in the prediction FE model. In the simplified HAWC2 models, this is acknowledged by restraining shaft rotation, disabling torsional deformations, and using stiff blades. The comparison aims at validating the effects of mass and stiffness terms, soil support conditions, and hydrodynamic mass used in the prediction FE model by gradually adding these terms. This yields the following three model setups for the simplified HAWC2 model:

- Model setup 1: Excluding the hydrodynamic elements and the soil model, and fixing the bottom node in all DOFs. This model resembles a bottom-fixed land-based wind turbine.

- Model setup 2: Excluding the hydrodynamic elements, while reintroducing the soil support from the original HAWC2 model in Section 2.1.

- Model setup 3: Introducing the hydrodynamic elements without water kinematics to reduce complexity.

The comparison of the natural frequencies of the simplified HAWC2 model $f_{n,HAWC2}$ and the prediction FE model $f_{n,Pred}$ are presented for the first seven modes in Table 5, in which the error is calculated as

$$\varepsilon(f_n) = \frac{f_{n,Pred} - f_{n,HAWC2}}{f_{n,HAWC2}} \tag{23}$$

As presented in Table 5, the error $\varepsilon(f_n)$ for the tower bending modes is within the range from $-0.88$ to $1.13\%$, while for the torsion mode the error range increases to $3.17 - 3.32\%$. The two models are created from different underlying beam theories and implemented in different software tools, whereby discrepancies are expected. Thus, the agreement in Table 5 is generally good, with the larger error for the torsion mode possibly arising from the geometric stiffness matrix $\mathbf{K}_{g,e}$ in (21) not affecting torsional deformations.

Based on the results in Table 5, it is concluded that the mass and stiffness terms and the soil model are reasonably implemented in the prediction FE model. Furthermore, the simple implementation of the hydrodynamic mass is deemed acceptable for the cases where waves and currents are not included in the analysis. However, it is acknowledged that the model cannot capture the effects of currents and waves, as well as boundary effects at the seabed and water line.

[Figure]

[Figure]

**Table 5.** Overview of comparison of natural frequencies of three different model setups for a simplified version of the IEA 15-MW RWT HAWC2 model and the prediction FE model presented in Section 3.3.

| Model setup | Mode No. | 1 | 2 | 3 | 4 | 5 | 6 | 7 |
|---|---|---|---|---|---|---|---|---|
| 1 | **Mode** | **1st bend.** | **1st bend.** | **2nd SS** | **2nd FA** | **1st torsion** | **3rd SS** | **3rd FA** |
| | $f_{n,HAWC2}$ | 1.31E-01 | 1.31E-01 | 6.79E-01 | 7.19E-01 | 8.05E-01 | 1.50E+00 | 1.61E+00 |
| | $f_{n,Pred}$ | 1.30E-01 | 1.31E-01 | 6.75E-01 | 7.12E-01 | 7.79E-01 | 1.52E+00 | 1.61E+00 |
| | $\varepsilon(f_n)$ | -0.55% | -0.16% | -0.48% | -0.88% | **-3.17%** | 1.13% | 0.06% |
| 2 | **Mode** | **1st bend.** | **1st bend.** | **1st torsion** | **2nd SS** | **2nd FA** | **3rd SS** | **3rd FA** |
| | $f_{n,HAWC2}$ | 1.61E-01 | 1.62E-01 | 8.01E-01 | 8.47E-01 | 9.15E-01 | 1.93E+00 | 2.02E+00 |
| | $f_{n,Pred}$ | 1.60E-01 | 1.61E-01 | 7.75E-01 | 8.52E-01 | 9.11E-01 | 1.95E+00 | 2.02E+00 |
| | $\varepsilon(f_n)$ | -0.80% | -0.29% | **-3.32%** | 0.54% | -0.47% | 0.94% | 0.16% |
| 3 | **Mode** | **1st SS** | **1st FA** | **1st torsion** | **2nd SS** | **2nd FA** | **3rd SS** | **3rd FA** |
| | $f_{n,HAWC2}$ | 1.61E-01 | 1.62E-01 | 8.01E-01 | 8.37E-01 | 9.00E-01 | 1.79E+00 | 1.87E+00 |
| | $f_{n,Pred}$ | 1.60E-01 | 1.61E-01 | 7.74E-01 | 8.41E-01 | 8.96E-01 | 1.81E+00 | 1.88E+00 |
| | $\varepsilon(f_n)$ | -0.83% | -0.49% | **-3.29%** | 0.43% | -0.47% | 0.97% | 0.22% |

**3.3.2 Ritz vectors**

As explained in Section 3.1, the predicted response $\mathbf{u}_p(t)$ of a dynamic system can be estimated as the sum of the predicted response in the individual frequency bands $B_i$ based on the mode shape matrix $\mathbf{\Phi}$. However, for large-scale OWTs, the quasi-static effects arising from e.g. yawing, wind, and waves significantly contribute to the response. These effects can be captured by a linear combination of higher-order modes. However, because a modal truncation omitting higher-order modes is needed in MDE, due to the limited number of sensors available, the accuracy of the predicted response may be compromised in the quasi-static region and between the resonant peaks. Different suggestions have been made to account for the quasi-static response, where Skafte et al. (2017) suggest the use of Ritz vectors, while similar methods are applied in Iliopoulos et al. (2017), Augustyn et al. (2021), and Toftekær et al. (2023). Furthermore, Tarpø (2020) compares the use of Ritz vectors with a *modal truncation augmentation method* and finds that the difference in performance is insignificant for the considered case. In the present work, the methodology using Ritz vectors based on static loads from Skafte et al. (2017) is applied, as explained in the following.

The mode shape matrix in (10) is extended to include not only the $n$ mode shapes of the dynamic system $\mathbf{\Phi}_d$ obtained from the eigenanalysis of the FE model presented in Section 3.3, but also the $m$ Ritz vectors obtained from static analysis $\mathbf{\Phi}_s$,

$$\mathbf{\Phi} = \begin{bmatrix} \mathbf{\Phi}_s \ \mathbf{\Phi}_d \end{bmatrix} \tag{24}$$

440    whereby $\boldsymbol{\Phi}$ becomes an $n_{dof} \times (m+n)$ array. The matrix $\boldsymbol{\Phi_s} = [\boldsymbol{\phi}_1, \boldsymbol{\phi}_2, \cdots, \boldsymbol{\phi}_m]$ contains the $m$ Ritz vectors ($\boldsymbol{\phi}_k$), obtained
by the static solution

$$\boldsymbol{\Phi}_s = \mathbf{K}^{-1}\mathbf{F} \tag{25}$$

where $\mathbf{K}$ is the stiffness matrix of the FE model presented in Figure 7 and $\mathbf{F}$ contains the static load vectors $\mathbf{f}_i$ representing
the load effects included in the MDE. Both Toftekær et al. (2023) and Iliopoulos et al. (2017) suggest that an appropriate Ritz
445    vector for the thrust load can be obtained by applying a horizontal nodal force at the top of the FE model tower, see Figure
9(a). Furthermore, Toftekær et al. (2023) show that a supplemental Ritz vector from the nodal tower-top moment in Figure
9(b) improves the MDE strain estimates associated with RNA yaw or uneven rotor loading. Finally, Skafte et al. (2017), Tarpø
(2020), and Augustyn et al. (2021) all include load from waves in the performed MDE, see Figure 9(c). In the present work,
three pairs of Ritz vectors are included in the MDE, representing the FA and SS directions, respectively. In each direction, the
450    tower-top nodal load (a) and moment (b), and the wave loading (c) are presented in Figure 9.

[Figure]

**Figure 9.** Loads and moments applied to determine the Ritz vectors for the estimation of the quasi-static response. Based on suggested loads
in Toftekær et al. (2023). (a) shows the tower-top nodal load, (b) shows the tower top moment, and (c) shows the wave loading.

The wave load depicted in Figure 9(c) is based on the expression for the total force

$$F_x(z,t) = \frac{2\rho g H}{k}\frac{\cosh(k\,(z+h))}{\cosh(kh)}\,A(kr_0)\cos(\omega t - \delta) \tag{26}$$

on a unit height of a vertical cylinder (Sumer and Fredsøe, 1997). In the present work, normalized displacements are used,
hence only the distribution across the water depth of the monopile is of interest, whereby the temporal and constant terms can

[Figure]

**Figure 10.** Ritz vectors in terms of displacement and bending moments extracted from the prediction FE model presented in Figure 7: (a) is based on nodal force in tower top, (b) is based on the nodal moment in the tower top, and (c) is based on the approximated wave load presented in (27). The three loads are illustrated in Figure 9.

455  be removed in (26). Thereby, the vertical distribution of the force (above the seabed) reduces to

$$F'_x(z) = \frac{\cosh(k\,(z+h))}{\cosh(kh)} \tag{27}$$

where $h = 30\,\text{m}$ is the water depth and $k = \frac{2\pi}{L}$ is the deep-water wave number, derived for the wave length $L = \frac{g}{2\pi}T^2$ with the wave period $T = 6.52\,\text{s}$ calculated for a hub wind speed of $V_{hub} = 10\,\text{m/s}$. The distributed force in (27) assumes that the wave loads are dominated by the inertia contribution in Morison's equation, while neglecting drag. This assumption is indeed

460  valid for $V_{hub} = 10\,\text{m/s}$, for which inertia forces constitute 98.5 % of the total force. However, extending the wave load Ritz vector to be wind speed dependent might be relevant, as suggested in Tarpø (2020). The Ritz vectors obtained from the load presented in Figure 9 are presented in Figure 10 in terms of displacements and bending moments.

**4   MDE estimation of damage equivalent loads and stresses**

The objective of the multi-band MDE is to obtain valid estimates of strains, stress, or force histories at any given location in

465  a given structure. The accuracy of the MDE depends not only on the quality of the FE model from Section 3.3, but also on the configuration and input data, which are presented in the next Section 4.1. The purpose of the applied multi-band Modal Decomposition and Expansion (MDE) is to evaluate the fatigue damage from bending stresses in any relevant location of the

supporting structure. Hence, the performance of the MDE should be assessed using a measure that accounts for the accuracy in terms of strains or forces, while also being consistent with how fatigue damage is evaluated. In Section 4.2 this comparison

470 is therefore conducted in terms of Damage Equivalent Loads (DELs) and Damage Equivalent Stresses (DESs).

**4.1 MDE setup**

This section presents the basis for the MDE performed for the IEA 15-MW RWT supporting structure in terms of sensor type and placement (i.e. the HAWC2 output channels in $\mathbf{u}_m(t)$), band separation used in the frequency domain, and the choices of Ritz vectors and mode shapes used within the individual bands ($\tilde{\mathbf{\Phi}}_i$).

475 As presented in Section 1, it is widely accepted in the literature that the dynamic part of the response $\mathbf{u}_p(t)$ can be predicted based on measured accelerations. From these accelerations, displacements are obtained through double integration. However, for the quasi-static part of the response, the displacements are often inaccurate because measurement noise in the acceleration measurements is amplified during low-frequency integration. To overcome this challenge, Iliopoulos et al. (2017) uses strain gauge measurements as input to the MDE for the quasi-static response estimation. Alternatively, Toftekær et al. (2023) uses the

480 low-pass filtered (vertical) accelerations obtained from DC accelerometers relative to the gravitational acceleration to estimate rotations. This has the advantage that no double integration must be performed, and no additional sensors must be installed. In the present work, $\mathbf{u}_m(t)$ therefore contains displacements and rotations for the prediction of dynamic and quasi-static responses, respectively (see Figure 11).

Obviously, the location of the accelerometers will impact the quality of the virtual sensors. Different methods have been used

485 to optimise the sensor placement (Mehrjoo et al., 2022; Ercan and Papadimitriou, 2021). However, in practical applications, accessibility is just as relevant for the installation of sensors, since maintenance and replacement of structural health monitoring systems play a central role in the robustness of the overall system. Thus, in the present work, the physical sensors are placed at locations where internal platforms are most likely installed inside the tower (see Figure 11).

As presented in Figure 6, the multi-band MDE (16) is performed by separating the response of the IEA 15-MW RWT

490 into four individual bands ($B_1$ to $B_4$) before combining them to the total predicted response $\mathbf{u}_p(t)$. This band separation captures the effects dominating the individual bands in terms of wind, waves, operational forces, or resonant responses without exceeding the inherent sensor limitations of the MDE. The justification of the present band separation is given below for the MDE configuration summarized in Table 6:

- $B_1$ represents the quasi-static domain of the response. The response in this frequency band is primarily driven by tur-

495 bulence. Thus, the Ritz vectors included for the prediction in this band are obtained from the nodal force and moment. Furthermore, the wind is assumed to act as a distributed load across the tower, whereby the first tower bending mode shapes are also included.

- $B_2$ represents the first dynamic band, governed by wave loading with a wave frequency of $1/T_p = 0.068$ Hz at $V = 35$ m/s and $1/T_p = 0.18$ Hz at $V = 4$ m/s. Furthermore, the wind load also contributes to the response in this frequency

500 band, whereby all three pairs of Ritz vectors are included.

[Figure]

[Figure]

**Figure 11.** Measurement locations i.e. HAWC2 output channels in red in terms of displacements $u_{m,*}(t)$ included in MDE in dynamic frequency range and rotations $\theta_{m,*}(t)$ included in MDE in quasi-static frequency range

- $B_3$ represents the second dynamic band, in which the first tower bending modes dominate the response along with the wave loads. Hence, the first tower bending mode shapes and the Ritz vectors from wave loading are included.

- $B_4$ includes the higher dynamics and rotor harmonics. Here, the first three pairs of tower bending modes are included, while the first tower torsion mode is omitted as it is considered less significant for estimating bending stresses.

505   The following section assesses the performance of the MDE using the configuration described above and the prediction FE model presented in Section 3.3. This is achieved by comparing DELs and DESs, calculated from section moment load histories, obtained from both the MDE and the true HAWC2 output time series. The comparison is performed in both the FA and SS directions and at all nodes in the supporting structure for the DLCs described in Section 2.3.

[Figure]

[Figure]

**Table 6.** Configuration used for MDE in the frequency ranges $B_1$, $B_2$, $B_3$, and $B_4$ in terms of measurements, mode shapes, and Ritz vectors.

| Band No. ($i$) | 1 | 2 | 3 | 4 |
|---|---|---|---|---|
| $B_i$ | $[0.00 - 0.05]$ Hz | $[0.05 - 0.13]$ Hz | $[0.13 - 0.25]$ Hz | $[0.25 - 50]$ Hz |
| $\mathbf{u}_{i,m}(t)$ | $\begin{bmatrix} \theta_1 \ \theta_2 \ \theta_3 \ \theta_4 \ \theta_5 \ \theta_6 \end{bmatrix}$ | $\begin{bmatrix} u_1 \ u_2 \ u_3 \ u_4 \ u_5 \ u_6 \end{bmatrix}$ | $\begin{bmatrix} u_1 \ u_2 \ u_3 \ u_4 \ u_5 \ u_6 \end{bmatrix}$ | $\begin{bmatrix} u_1 \ u_2 \ u_3 \ u_4 \ u_5 \ u_6 \end{bmatrix}$ |
| $\tilde{\boldsymbol{\Phi}}_{i,s}$ | $\begin{bmatrix} \phi_1 \ \phi_2 \ \phi_3 \ \phi_4 \end{bmatrix}$ | $\begin{bmatrix} \phi_1 \ \phi_2 \ \phi_3 \ \phi_4 \ \phi_5 \ \phi_6 \end{bmatrix}$ | $\begin{bmatrix} \phi_5 \ \phi_6 \end{bmatrix}$ | - |
| $\tilde{\boldsymbol{\Phi}}_{i,d}$ | $\begin{bmatrix} \varphi_1 \ \varphi_2 \end{bmatrix}$ | - | $\begin{bmatrix} \varphi_1 \ \varphi_2 \end{bmatrix}$ | $\begin{bmatrix} \varphi_1 \ \varphi_2 \ \varphi_4 \ \varphi_5 \ \varphi_6 \ \varphi_7 \end{bmatrix}$ |

**4.2 Damage equivalent loads and stresses**

510 Fatigue Damage Equivalent Loads (DELs) reduce a load history to a single equivalent load range $\Delta P_{eq}$, which is defined as the constant amplitude 1 Hz sinusoidal load causing the same amount of fatigue damage as the original load history. The same applies for fatigue Damage Equivalent Stresses (DESs) $\Delta S_{eq}$, making DELs and DESs convenient measures for comparing fatigue contributions across load cases with different durations (Veldkamp, 2006). Thus, in the present section, the DELs and DESs combined for the individual DLCs presented in Section 2.3 are compared and discussed. Furthermore, the MDE

515 performance is assessed, initially for DELs and DESs calculated for the individual DLCs and subsequently for the DESs calculated for the individual HAWC2 section moment time histories. In both cases, the comparison is performed in all nodes of the IEA 15-MW RWT HAWC2 model.

The DEL for a single load history $\Delta P_{eq,s}$ can be calculated as in (6), where $n_{eq}$ is the number of 1 Hz cycles in the considered time series. Similarly, the DEL for the individual DLCs can be calculated as

$$\quad \Delta P_{eq,DLC} = \left( \frac{\sum_{s \in \text{DLC}} n_{eq}(\Delta P_{eq,s})^m}{n_{eq,DLC}} \right)^{\frac{1}{m}} \tag{28}$$

where

$$n_{eq,DLC} = n_{eq}\, n_{seed,DLC} \tag{29}$$

is the total number of 1 Hz cycles in the simulations contained in the individual DLCs, with $n_{seed,DLC}$ being the simulation seeds for the individual DLC (i.e., the number of (converged) simulations in Table 2 for a given DLC at MWL equal to MSL).

525 Inserting (29) in (28) yields the more compact representation

$$\Delta P_{eq,\text{DLC}} = \left( \frac{\sum_{s \in \text{DLC}}(\Delta P_{eq,s})^m}{n_{seed,DLC}} \right)^{\frac{1}{m}} \tag{30}$$

As the DEL retains the unit of load, the DES $\Delta S_{eq,s}$ can be obtained by applying Navier's stress distribution formula to the DEL $\Delta P_{eq,s}$ for the individual nodes of interest in the supporting structure. However, the elements in the IEA 15-MW RWT are not consistent in terms of bending stiffness across the nodes, whereby Navier's formula will produce discontinuous stresses

530 at the nodes. Thus, only the DES associated with the maximum nodal stresses in the monopile and tower circumference are considered for each node. Furthermore, only the contributions arising from the bending moments are included in the DESs

[Figure]

**Figure 12.** DELs calculated for the individual DLCs based on section moment load histories from HAWC2 (——) and MDE prediction (······) in the FA (a) and SS (b) direction of the IEA 15-MW RWT, as presented in (30).

which are calculated as

$$\Delta S_{eq,\text{DLC}} = \left( \frac{\sum_{s \in \text{DLC}} (\Delta S_{eq,s})^m}{n_{seed,DLC}} \right)^{\frac{1}{m}} \tag{31}$$

for the individual DLCs.

535    Figures 12 and 13 show the DELs and DESs related to the FA and SS section moments obtained from the HAWC2 simulations directly (——) and predicted using the multi-band MDE configuration from Section 4.1 (······).

[Figure]

**Figure 13.** DESs calculated for the individual DLCs based on section moment load histories from HAWC2 (——) and MDE prediction (······) in the FA (a) and SS (b) direction of the IEA 15-MW RWT, as presented in (31).

As illustrated in Figure 12, the DELs generally look similar to the moment curve from the first tower bending modes or the thrust load (see Figure 8 and 10), with overlying effects from other loads and modes. In the FA direction (a), the operating DLCs 1.2 and 2.4 generally induce lower DELs compared to DLCs 3.1, 4.1, 6.4, and 7.2, with DLC 6.4 resulting in the maximum DEL across all DLCs and directions (FA and SS) at the mud line. The lower DELs of DLCs 1.2 and 2.4 can be attributed to the significant aerodynamic damping provided by the operating rotor, as discussed in Section 2.5. However, within the tower top region, specifically from around $120 - 144\,\text{m}$, the operating DLCs show higher DELs due to uneven loading of the rotor and 3P effects, as discussed in Section 2.5. In the SS direction (b), in which the aerodynamic damping, the effects from thrust

load variations, and the 3P effects have less influence, the differences in DEL between operating and non-operating DLCs are

545  generally smaller than those observed in the FA direction. It is worth noting that DLC 7.2 results in significantly higher DELs than all other DLCs at elevations above approximately 75 m. This can be attributed to the excitation of the second tower SS mode and the blade vibrations specific to this DLC, as described in further detail in Section 2.5.

In Figure 12, it is observed that for all DLCs in both the FA and SS directions, the MDE underestimates the DELs in a $\pm 15$ m zone around the MSL. Because the error occurs in both the FA and SS direction, it is not expected to derive from inadequate

550  modelling of the wave load. Instead, it is most likely caused by not representing the rotor flexibility in the second tower bending modes, which have a great impact on the DEL at the present location.

An inherent problem of the DELs in Figure 12 is that they do not explicitly account for changes in cross-section dimensions, whereby small DELs might still cause large stresses in regions with small tower diameters. Thus, in Figure 13, the DESs have large values in the tower-top region, where the corresponding DELs in Figure 12 are small. This indicates that the accuracy of

555  the MDE cannot be ignored in the tower-top region. For the present analysis in Figure 13, this is especially important for DLCs 1.2 and 2.4 in the FA direction (a), and DLC 7.2 in the SS direction (b), which have their DES maxima in the tower-top region.

For the DESs estimated by MDE in Figure 13, it is seen that the multi-band MDE performs poorly at the tower top, where it consistently underestimates the DESs in the FA direction, and significantly overestimates the DESs in the SS direction for DLC 1.2 and 2.4. As discussed in section 2.5, the damage in the tower top is governed mostly by different phenomena associated

560  with the rotor and blade dynamics, which are omitted in the RNA model. This may be the root cause of the large deviations observed for the DESs.

The MDE performance discussed above and presented in Figures 12 and 13 is based on a combined DEL and DES calculated for the individual DLCs for each elevation $z$ along the IEA 15-MW RWT supporting structure. Thus, it corresponds to an averaged or mean error, conveniently used for assessing long-term MDE performance, although inherently sensitive to bias

565  errors. Therefore, to assess the short-term performance of the MDE in the individual HAWC2 simulations, the relative error of the DESs is calculated for the individual HAWC2 simulations as

$$\varepsilon_{MDE} = \frac{\Delta S_{eq,s,MDE}}{\Delta S_{eq,s,HAWC2}} - 1 \tag{32}$$

where $(\ )_{HAWC2}$ denotes the DESs calculated from the HAWC2 time series of the FA and SS section moments and $(\ )_{MDE}$ denotes the DESs calculated from the corresponding MDE estimate. Figure 14 presents the relative error $\varepsilon_{MDE}$ of the DESs,

570  related to the FA and SS section moment and calculated for each elevation $z$ along the IEA 15-MW RWT supporting structure.

[Figure]

[Figure]

**Figure 14.** Error $\varepsilon_{MDE}$ of DESs for the MDE predicted section moment load histories in the FA (top) and SS (bottom) direction of the IEA 15-MW RWT from the individual HAWC2 simulation $s$, as presented in (32). Color gradient represents the mean wind speed at the hub $V_{hub}$ for the considered simulation $s$.

It is observed in Figure 14 that the error $\varepsilon_{MDE}$ is predominantly in the range of $\pm 5\%$, except at the tower top, where the MDE performs inconsistently for the various DLCs. The error generally shows a dependency on the wind speed, which can be attributed to the operational and environmental variability of the IEA 15-MW RWT, arising from the varying rotor speeds, changing turbulence, and changing wave loads, which cannot be captured by the MDE, assuming a linear and time-invariant

575   response.

[Figure]

[Figure]

In the FA direction at the tower top elevation from $135 - 144\,\mathrm{m}$ in Figure 14(top), the error $\varepsilon_{MDE}$ appears to be inversely proportional to the wind speed for DLCs 1.2 and 2.4. As previously mentioned, the 3P effects significantly influence the DELs in the tower top for the operating load cases. However, the 3P effects include both tower shadow effects, wind shear, and turbulence, which makes it wind speed dependent. Therefore, the tower shadow effects can dominate in the low wind speed regime, while turbulence takes over at higher wind speeds, thereby modifying the response characteristics and consequently the MDE prediction accuracy. For DLC 6.4, no wind speed dependency of the MDE error is observed at the tower top. This is expected, as the tower-top DESs for this DLC are mainly governed by the inherent dynamics of the wind turbine (first tower FA mode and first edgewise blade mode), which are not influenced by operational variability (e.g., gyroscopic stiffening and blade pitching) in idle conditions. A similar response could then be expected for DLC 7.2 (also at standstill), where a large variance is however observed for the MDE error at the tower top. The difference between DLC 7.2 and 6.4 is the locked rotor configuration, which therefore must be the main cause of the MDE's inability to represent the tower top response from DLC 7.2, while the different azimuth angles of the locked rotor for this DLC can also affect the variance of tower top moment.

For the SS response, the error $\varepsilon_{MDE}$ at the tower top in Figure 14(bottom) exhibits a high variability that appears proportional to the wind speed for DLCs 1.2 and 2.4. Because a similar error pattern is not observed for the non-operating DLC and the error for DLCs 1.2 and 2.4 highly depends on the wind speed, it is concluded that the error is related to the effects from the operating rotor, not captured by the MDE.

For DLC 7.2, it is observed that the MDE tends to underestimate the DES for low wind speeds while overestimating it for higher wind speeds. Furthermore, the error increases to a range between $\pm 25\,\%$, which is somewhat surprising considering the low discrepancies between DELs and DESs calculated from the MDE estimates and the HAWC2 outputs in Figures 12 and 13. Because the blade dynamics and second tower SS mode are significant at the tower top for DLC 7.2, it is concluded that the error is related to the too simple modelling of the RNA in the prediction FE model. Conversely, the wind speed dependency is more challenging to assess, although the associated increase in wind turbulence excites different modes.

Finally, the error $\varepsilon_{MDE}$ between the MSL and the mud line in Figure 14 depends on the wind speed for all DLCs in the FA direction, while less so in the SS direction, as most clearly seen for DLC 6.4. This discrepancy is likely an effect of how the Ritz vectors include wave loads, i.e. not accounting for their sensitivity to wave height fluctuations or the dynamic interchange between drag and inertia forces. Furthermore, the wave load is applied to the monopile between mud line and MSL, thus ignoring the change in loading area during the transition from wave top to crest. In conclusion, the wave load Ritz vector is unable to capture the full complexity of the actual wave load in the IEA 15-MW RWT HAWC2 model.

When combining the conclusions from the above discussion, it is assessed that the MDE used in the present work generally performs well, except at the tower top. Hereby, the main challenges associated with the present use of MDE are:

- Capturing the local effects of the flexible and dynamic response of the rotor and blades.

- Including the effects from rotor flexibility and operation in the tower mode shapes used in the MDE.

- Including wind speed variability and time dependency of the waves in the MDE.

[Figure]

[Figure]

Some of the errors observed in the present section may also be related to the chosen sensor locations and the associated MDE
610 configuration presented in Section 4.1. However, as noise is not included in the present analysis, the noise-to-signal ratio is not
an issue, whereby a non-optimal sensor location would have less impact in the present comparison.

**5 Conclusions**

This paper presents an overview of the dataset available in Pedersen et al. (2025), containing response simulations covering
the Fatigue Limit State (FLS) design life of the IEA Wind 15-Megawatt Offshore Reference Wind Turbine with a monopile
615 foundation (IEA 15-MW RWT) version 1.1.6.

The paper explores how diverse operational and environmental scenarios impact the Damage Equivalent Loads (DELs) cal-
culated from the Fore-Aft (FA) and Side-Side (SS) section moment histories at the tower base, after which the relative lifetime
damage for the individual FLS Design Load Cases (DLCs), described in IEC 61400-3-1:2019 (IEC, 2019b), is calculated at
all nodes in the supporting structure of the IEA 15-MW RWT. It has been found that the DLCs representing *power production
in normal conditions* (DLC 1.2), *parked turbine with idle rotor in normal conditions* (DLC 6.4), and *fault - locked rotor in
normal conditions* (DLC 7.2) govern the lifetime damage of the supporting structure. The high contribution from DLC 1.2
occurs because of its high duration (90% of the design life) and the excitation at the tower top caused by 3P effects, while the
contribution of DLCs 6.4 and 7.2 is large because of their high DELs associated with low aerodynamic damping.

The paper gives an overview of multi-band Modal Decomposition and Expansion (MDE) and a methodology for expressing
625 the estimated response in sectional forces, after which it presents the Finite Element (FE) model used to calculate the Ritz
vectors and mode shapes used to perform MDE. It explains the configuration used to perform MDE for the estimation of section
moment time histories in the supporting structure of the IEA 15-MW RWT, which is based on rotation and displacement data
from six HAWC2 sensors located at three elevations in the RWT tower (in both the FA and SS direction), and includes both
the quasi-static and dynamic part of the frequency response.

630 The present work utilises MDE to estimate section moment histories in all nodes of the supporting structure of the IEA 15-
MW RWT across different operational and environmental regimes represented in the data from Pedersen et al. (2025). Based
on the moment histories, the combined DELs of the individual DLCs are calculated along with the combined DESs for the
individual DLCs and the DESs from the individual HAWC2 simulations. The MDE generally performs well in estimating the
combined DELs and DESs for the individual DLCs. However, notable errors occur around the tower top, specifically from 120
635 - 144 m above the Mean Sea Level (MSL), and at the MSL $\pm 15$ m. These errors are attributed to the omission of local effects in
the blade dynamics, and to blade flexibility not being included in the second tower bending mode shapes when using a lumped
inertia Rotor-Nacelle-Assembly (RNA) model. The relative MDE errors for the DESs of the individual HAWC2 simulations
$\varepsilon_{MDE}$ are predominantly in the range of $\pm 5\%$, thus confirming that the MDE performs well in general. These MDE errors
also underline that the MDE performs poorly around the tower top, where errors up to $180\%$ are observed. Finally, the MDE
640 errors show a wind speed dependency, except in the SS direction, when the rotor idle. It is concluded that the wind speed
dependency of the MDE error is caused by environmental and operational variability of the rotor, which is not captured by

the MDE assuming a linear and time-invariant response. Additionally, the lumped inertia RNA model and the wave load Ritz vector, which do not incorporate wind speed variability and the time-dependent nature of waves, likely contribute further to the observed wind speed dependency of the MDE error.

645    In future work, the authors suggest investigating errors in the frequency domain to increase confidence in the observed causes of error. The knowledge obtained from the present work will serve as a basis for updating the RNA model to include blade flexibility, and subsequently to include operational and environmental variability in the RNA modelling, e.g. by using individual RNA models for various wind speeds. The authors also plan to implement a wave load model that accounts for the waves' variation with the wind speed. Finally, it would be vital to investigate the MDE accuracy of a reduced number of

650    physical sensors, e.g. from existing monitoring systems, not specifically designed for virtual sensing purposes.

[Figure]

*Data availability.*  Dataset with synthetic wind turbine response data is available at https://doi.org/10.11583/DTU.24460090.

*Code and data availability.*  Python code for reading data is available at https://github.com/madg-DTU/IEA-15MW-RWT-HAWC2-Monopile-Response-Database

[Figure]

**Appendix A: Properties of IEA 15-MW RWT supporting structure**

**Table A1.** Structural properties of element $e$ in the IEA 15-MW RWT supporting structure. Including the node coordinates of the end nodes in the element $n_{e,1}$ and $n_{e,2}$, the Young's modulus $E$, the shear modulus $G$, the outer radius $r$, the cross section area $A$, the moments of inertia $I_{xx}$ and $I_{yy}$, the polar moment of inertia $I_p$, and the distributed mass $m$ along the height $z$.

| Element No. | Coord $n_{e,1}$ [m] | Coord. $n_{e,2}$ [m] | $E$ [Pa] | $G$ [Pa] | $r$ [m] | $A$ [m$^2$] | $I_{xx}$ [m$^4$] | $I_{yy}$ [m$^4$] | $I_p$ [m$^4$] | $m$ [kg/m] |
|---|---|---|---|---|---|---|---|---|---|---|
| 1 | $(0,0,-75)$ | $(0,0,-70)$ | 2.00E+11 | 7.93E+10 | 5.00E+00 | 1.73E+00 | 2.14E+01 | 2.14E+01 | 4.27E+01 | 1.44E+04 |
| 2 | $(0,0,-70)$ | $(0,0,-65)$ | 2.00E+11 | 7.93E+10 | 5.00E+00 | 1.73E+00 | 2.14E+01 | 2.14E+01 | 4.27E+01 | 1.44E+04 |
| 3 | $(0,0,-65)$ | $(0,0,-60)$ | 2.00E+11 | 7.93E+10 | 5.00E+00 | 1.73E+00 | 2.14E+01 | 2.14E+01 | 4.27E+01 | 1.44E+04 |
| 4 | $(0,0,-60)$ | $(0,0,-55)$ | 2.00E+11 | 7.93E+10 | 5.00E+00 | 1.73E+00 | 2.14E+01 | 2.14E+01 | 4.27E+01 | 1.44E+04 |
| 5 | $(0,0,-55)$ | $(0,0,-50)$ | 2.00E+11 | 7.93E+10 | 5.00E+00 | 1.73E+00 | 2.14E+01 | 2.14E+01 | 4.27E+01 | 1.44E+04 |
| 6 | $(0,0,-50)$ | $(0,0,-45)$ | 2.00E+11 | 7.93E+10 | 5.00E+00 | 1.73E+00 | 2.14E+01 | 2.14E+01 | 4.27E+01 | 1.44E+04 |
| 7 | $(0,0,-45)$ | $(0,0,-40)$ | 2.00E+11 | 7.93E+10 | 5.00E+00 | 1.73E+00 | 2.14E+01 | 2.14E+01 | 4.27E+01 | 1.44E+04 |
| 8 | $(0,0,-40)$ | $(0,0,-35)$ | 2.00E+11 | 7.93E+10 | 5.00E+00 | 1.73E+00 | 2.14E+01 | 2.14E+01 | 4.27E+01 | 1.44E+04 |
| 9 | $(0,0,-35)$ | $(0,0,-30)$ | 2.00E+11 | 7.93E+10 | 5.00E+00 | 1.73E+00 | 2.14E+01 | 2.14E+01 | 4.27E+01 | 1.44E+04 |
| 10 | $(0,0,-30)$ | $(0,0,-25)$ | 2.00E+11 | 7.93E+10 | 5.00E+00 | 1.73E+00 | 2.14E+01 | 2.14E+01 | 4.27E+01 | 1.44E+04 |
| 11 | $(0,0,-25)$ | $(0,0,-20)$ | 2.00E+11 | 7.93E+10 | 5.00E+00 | 1.67E+00 | 2.07E+01 | 2.07E+01 | 4.13E+01 | 1.39E+04 |
| 12 | $(0,0,-20)$ | $(0,0,-15)$ | 2.00E+11 | 7.93E+10 | 5.00E+00 | 1.61E+00 | 1.99E+01 | 1.99E+01 | 3.98E+01 | 1.34E+04 |
| 13 | $(0,0,-15)$ | $(0,0,-10)$ | 2.00E+11 | 7.93E+10 | 5.00E+00 | 1.55E+00 | 1.92E+01 | 1.92E+01 | 3.83E+01 | 1.29E+04 |
| 14 | $(0,0,-10)$ | $(0,0,-5)$ | 2.00E+11 | 7.93E+10 | 5.00E+00 | 1.49E+00 | 1.84E+01 | 1.84E+01 | 3.68E+01 | 1.24E+04 |
| 15 | $(0,0,-5)$ | $(0,0,0)$ | 2.00E+11 | 7.93E+10 | 5.00E+00 | 1.42E+00 | 1.76E+01 | 1.76E+01 | 3.53E+01 | 1.19E+04 |
| 16 | $(0,0,0)$ | $(0,0,5)$ | 2.00E+11 | 7.93E+10 | 5.00E+00 | 1.36E+00 | 1.69E+01 | 1.69E+01 | 3.37E+01 | 1.14E+04 |
| 17 | $(0,0,5)$ | $(0,0,10)$ | 2.00E+11 | 7.93E+10 | 5.00E+00 | 1.32E+00 | 1.64E+01 | 1.64E+01 | 3.28E+01 | 1.10E+04 |
| 18 | $(0,0,10)$ | $(0,0,15)$ | 2.00E+11 | 7.93E+10 | 5.00E+00 | 1.28E+00 | 1.59E+01 | 1.59E+01 | 3.19E+01 | 1.07E+04 |
| 19 | $(0,0,15)$ | $(0,0,30)$ | 2.00E+11 | 7.93E+10 | 5.00E+00 | 1.22E+00 | 1.52E+01 | 1.52E+01 | 3.03E+01 | 1.01E+04 |
| 20 | $(0,0,30)$ | $(0,0,45)$ | 2.00E+11 | 7.93E+10 | 4.99E+00 | 1.11E+00 | 1.36E+01 | 1.36E+01 | 2.72E+01 | 9.22E+03 |
| 21 | $(0,0,45)$ | $(0,0,60)$ | 2.00E+11 | 7.93E+10 | 4.89E+00 | 9.85E-01 | 1.10E+01 | 1.10E+01 | 2.21E+01 | 8.18E+03 |
| 22 | $(0,0,60)$ | $(0,0,75)$ | 2.00E+11 | 7.93E+10 | 4.58E+00 | 8.65E-01 | 8.36E+00 | 8.36E+00 | 1.67E+01 | 7.20E+03 |
| 23 | $(0,0,75)$ | $(0,0,90)$ | 2.00E+11 | 7.93E+10 | 4.21E+00 | 7.42E-01 | 5.95E+00 | 5.95E+00 | 1.19E+01 | 6.18E+03 |
| 24 | $(0,0,90)$ | $(0,0,105)$ | 2.00E+11 | 7.93E+10 | 3.78E+00 | 6.25E-01 | 4.07E+00 | 4.07E+00 | 8.14E+00 | 5.20E+03 |
| 25 | $(0,0,105)$ | $(0,0,120)$ | 2.00E+11 | 7.93E+10 | 3.47E+00 | 5.13E-01 | 2.98E+00 | 2.98E+00 | 5.95E+00 | 4.28E+03 |
| 26 | $(0,0,120)$ | $(0,0,135)$ | 2.00E+11 | 7.93E+10 | 3.37E+00 | 4.46E-01 | 2.44E+00 | 2.44E+00 | 4.87E+00 | 3.72E+03 |
| 27 | $(0,0,135)$ | $(0,0,144)$ | 2.00E+11 | 7.93E+10 | 3.28E+00 | 4.90E-01 | 2.59E+00 | 2.59E+00 | 5.18E+00 | 4.09E+03 |

655 *Author contributions.* Conceptualization and methodology: MGP, JR, IFA, and JH; Wind turbine response simulations: MGP and JR; Data preparation and interpretation: MGP and JR; Prediction FE model: MGP and JH; Modal decomposition and expansion: MGP; writing (original draft): MGP; supervision and writing (review and editing): JR, IFA, and JH

[Figure]

*Competing interests.* The contact author has declared that none of the authors has any competing interests.

*Acknowledgements.* This work is partially funded by the Innovation Fund Denmark (grant 1155-00008B) and COWIfonden (grant C-
660 153.01). The authors acknowledge the use of AI language models for proofreading and enhancing the readability of this manuscript.

---

## Author Comment (AC1)

**Reviewer 1**

**RC1.01)** Clarify the novelty of your approach compared to existing studies, highlighting specific contributions and advances.

**RC1.01)** Thank you for pointing this out. The novel contributions of this paper are summarised below:

- This work shows how the structure-wide performance of multi-band MDE is limited by the lumped inertia RNA model. It specifically shows how errors are associated with erroneous second and third tower bending modes, given the omission of rotor modes coupled to tower excitation.
- This work proposes using a simple time-invariant load distribution for the wave load Ritz vector, which has not been explicitly defined in the considered existing studies dealing with OWTs on monopile foundations.
- Finally, this work is the first to utilise the comprehensive dataset from the IEA-15-MW-RWT-Monopile Database Pedersen et al. (2025). This dataset (1) facilitates cross-institute benchmarking of virtual sensing algorithms, as it provides an unrestricted range of sensor locations and quantities, and (2) enables validation of predicted response in the complete OWT, including monopile tower and blades.

This justification has been added to the paper as (line 84-91):

"The novel contributions of this paper are summarised as follows. This paper demonstrates how the structure-wide performance of multi-band MDE is limited by the lumped inertia RNA model. It specifically shows how errors are associated with erroneous second and third tower bending modes and the omission of rotor modes coupled to tower excitation. Additionally, this paper proposes a simple time-invariant load distribution for the wave load Ritz vector, which, to the best of the author's knowledge, has not been explicitly defined in existing studies dealing with virtual sensing in OWTs on monopile foundations. Finally, this is the first work to utilise the dataset (Pedersen et al., 2025). This dataset facilitates cross-institute benchmarking of virtual sensing algorithms, as it provides an unrestricted range of sensor locations and associated output channels. Furthermore, it enables validation of the predicted response in the entire OWT, including the monopile, tower, and blades."

**RC1.02)** Consider shortening the Data section, moving detailed information to an appendix to maintain focus on virtual sensing

**RC1.02)** Thank you for the comment. The authors recognise that the data section is long. To meet this point of critique, the Data section has been split into two sections: "2 Data" (line 99-110), which briefly lists the key information regarding the dataset (Pedersen et al., 2025), and "3 IEA 15-MW RWT performance and relative damage assessment" (line 111-229), which collects the former sections "2.4 Performance of the IEA 15-MW RWT" and "2.5 Relative lifetime damage results". The authors are of the opinion that section 3 is necessary context to fully interpret the MDE results and support the paper's conclusions. However, the former sections "2.1 IEA Wind 15-Megawatt Offshore Reference Wind Turbine", "2.2 Modelling", and "2.3 Load cases" have been collected in Appendix A.

**RC1.03)** Provide a rationale for the multi-band approach boundaries in Table 6, indicating if they are standard or proposed by the authors.

**RC1.03)** Thank you for this suggestion, it has resulted in significant improvements in the revised manuscript. A rationale for the frequency bands' boundaries has been added to the paper, and Figure 11 (previously 6), presenting the bands, has been moved to this section of the paper.

Furthermore, based on this comment and RC1.05 and RC1.06, we looked into the band separation and performed an extensive analysis. Based on the results, we changed the boundaries of B1 to include the 3P effects in this band and represent this with the Ritz vector obtained from the nodal moment. The altered text in the paper is presented here (line 434-459): "The rationale for the band separation depends on case-specific factors, including the frequency distribution of the external loads, the dynamic properties of the considered structure, and the properties of the sensors available in the monitoring system. Thus, the frequency bands should be selected such that the response is predicted accurately without exceeding the inherent sensor limitations of the MDE. The justification of the present band separation is given below for the MDE configuration summarised in Table 3:

– B1 is defined with an upper limit of 0.05 Hz. According to Toftekær et al. (2023), accurate displacements cannot be obtained from measured accelerations at frequencies below 0.05 Hz. Hence, the measured DOFs in $\Phi_m$ are defined in terms of rotations in B1, and the boundary represents a practical limitation of the sensors. B1 represents the quasi-static domain of the response, primarily driven by turbulence. Thus, the Ritz vectors included for the prediction in this band are obtained from the nodal force and moment in Figure 8(a,b). Furthermore, the wind is assumed to act as a distributed load across the tower, whereby the first tower bending mode shapes in Figure 7(a) are also included in the MDE.

– B2 is defined within the frequency range 0.05 to 0.13 Hz. The upper limit is chosen as the boundary between the thrust-dominated and the resonant parts of the response, dominated by the first tower bending modes. B2 is governed by wave loading with a wave frequency of $1/T_p$ = 0.068 Hz at V = 35 m/s and $1/T_p$ = 0.18 Hz at V = 4 m/s for the given site conditions. Furthermore, the wind load also contributes significantly to the response in this frequency band, whereby all three pairs of Ritz vectors in Figure 9 are included in the MDE for this band.

– B3 is defined within the frequency range 0.13 to 0.45 Hz. The upper limit is defined as the boundary between the 3P frequency and the frequency of the first flapwise blade mode. B3 is governed by the first tower bending modes along with the wave loads and the 3P excitation. Hence, the first tower bending mode shapes in Figure 7(a) and the Ritz vectors from wave loading in Figure 8(c) are included in the MDE. As the 3P excitation is driven primarily by uneven thrust loading on the rotor, it is well represented by the Ritz vector obtained from a nodal moment in Figure 8(b), hence, the Ritz vector in Figure 9(b) is also included in B3 for the MDE.

– B4 is defined within the frequency range 0.45 to 50 Hz. This frequency band represents a part of the response where the external loads are of minor influence. Hence, B4 includes the higher-order dynamics and rotor harmonics. Here, the first three pairs of tower bending modes in Figure 7 are included in the MDE, while the first tower torsion mode is omitted as it is considered less significant for estimating bending stresses."

**RC1.04)** Consider conducting a direct comparison between models with and without accurately modelled RNA, including rotor blades, to isolate error sources.
**RC1.04)** We thank the reviewer for this comment. The authors agree that including an accurately modelled RNA would indeed help to isolate the cause of the MDE error, and the authors intend to pursue this in future research. This is highlighted in the text in the conclusion (line 622-625):
"In future work, the authors suggest investigating the effects of including a flexible rotor in the FE model used to obtain the mode shapes used in the MDE. The knowledge obtained from the present work will serve as a basis for updating the RNA model to include blade flexibility, and

subsequently to include operational and environmental variability in the RNA modelling, e.g. by using individual RNA models for various wind speeds."

However, the authors consider this to be beyond the scope of the present paper. With the improvements made to the paper by implementing the revisions based on RC1.05 and RC1.06, the authors believe that the present paper provides sufficient argumentation for why a flexible rotor model should be included in the prediction FE model.

**RC1.05)** Offer a sample time series of strain data and analyse where discrepancies originate, enhancing the understanding of the study's context.

**RC1.06)** Investigate errors in the frequency domain to offer deeper insights into their origins and behaviour.

**RC1.05+RC1.06)** Thank you for this good suggestion for improving our work. To meet these points of critique, the following changes/additions have been made to the paper:

1) An Appendix C with selected sample moment time histories and their associated normalised power spectral density (PSD) has been added to the paper.

2) The PSD plots in Appendix C also include the PSD of the MDE error, the wind speed and the wave amplitude for a more qualified interpretation of the errors shown in the MDE error plots (Figures 12, 13, and 14).

3) The discussion in the results Section 5.2 (previously 4.2) (line 490-585) has been adjusted to include the results shown in the appendix.

4) The conclusions regarding the effects of disregarding blade flexibility from the FE model used in the MDE have been backed up by a statement in the introduction (line 66-70):
"This is demonstrated by (Reinhardt et al., 2024) which shows that ignoring blade flexibility in the RNA model significantly impacts the natural frequency and mode shape of the second tower bending modes. Additionally, rotor modes, which, given the inherent coupling between the tower and the blades, also affect the tower vibrations, are omitted from the MDE, as these cannot be represented using a lumped inertia RNA model. These simplifications can therefore introduce errors in the strains or stresses estimated in the supporting structure."

**References**

Pedersen, M. G., Rinker, J., Høgsberg, J., & Farreras, I. A. (2025). *IEA-15MW-RWT-Monopile HAWC2 Response Database*. Technical University of Denmark. https://doi.org/10.11583/DTU.24460090.v3

Reinhardt, T., Sastre Jurado, C., Weijtjens, W., & Devriendt, C. (2024). On the influence of rotor nacelle assembly modelling on the computed eigenfrequencies of offshore wind turbines. *Journal of Physics: Conference Series*, *2767*(5). https://doi.org/10.1088/1742-6596/2767/5/052034

---

## Author Comment (AC2)

**Reviewer 2**

**RC2.01) (line 23)** In my opinion, a very good abstract manages to communicate the following things:

- Give the context of the present work

- Point to the research question(s)

- Describe the methods

- Mention the most important results

All of which ideally should be done in about 150 words and without the use of acronyms.

If you agree with above criteria, I would then say the present abstract is a tad too long.

As one example. the entire description of the simulations is very detailed and does not necessarily have to ben in the abstract. Instead of giving the code, the type of DLCs and the fact it is 10-minute time series, the first sentence of the second paragraph can almost cover everything you need to mention in an abstract.

That being said, I am not the ultimate expert on the matter and if you say there are so many works out there which use 8 minute time-series and just DLC3.1 and 4.1, than leave everything as is and ignore this comment.

**RC2.01)** Thank you for this comment. We agree that the abstract was too long. The abstract has been rewritten to approximately 200 words, however, the MDE acronym has been kept as the authors believe it improves the readability (line 1-12):

**Abstract.** Offshore Wind Turbines are increasingly susceptible to fatigue damage, motivating structure-wide stress monitoring for asset integrity management and life extension. Virtual sensing methodologies, such as multi-band Modal Decomposition and Expansion (MDE), offer a solution to the above by extrapolating measurements from a few sensors at accessible locations to the global structure. However, most MDE studies model the Rotor-Nacelle-Assembly as a lumped mass inertia, thereby ignoring rotor flexibility. This leads to errors in estimated strains or stresses arising from erroneous mode shapes and the omission of relevant rotor modes from the estimates. The present paper quantifies these errors using HAWC2 simulations of the IEA 15-MW Offshore Reference Wind Turbine (RWT). Multi-band MDE estimates of section moments are compared to true responses in terms of Damage Equivalent Loads and Stresses. Long-term estimates show reduced accuracy in the area around the tower top and at ± 15 m around the Mean Sea Level. Furthermore, the error of the MDE estimates exhibits wind speed dependency, which underlines the inherent limitation of the MDE, assuming a linear and time-invariant response. In conclusion, multi-band MDE provides accurate estimates of section moments across most of the IEA 15-MW RWT supporting structure, although improvements are needed to effectively capture the influence from rotor flexibility.

**RC2.02) (line 27)** The V236 is already installed at least as a prototype and chinese OWT have reached almost 300 m with prototypes (MingYang 20 MW).

**RC2.02)** Thank you for pointing this out. The section has been edited to reflect the comment (line14-18):

"During recent decades, wind turbines have been consistently growing in size, and modern Offshore Wind Turbines (OWTs) already on the market, such as the Vestas V236-15MW, now

have a power production of up to 15 MW and rotor diameters approaching 240 m (Vestas Wind Systems A/S). At the same time, prototypes of the Mingyang MySE18.X-20MW, with a power production of 20 MW and a rotor diameter of up to 292 m, and the Siemens Gamesa SG DD-276, with a power production of 21.5 MW and a rotor diameter of up to 276 m, have also been installed (Ghoshal, 2024; Salas, 2025)."

**RC2.03) (line 30)** This is just a style issue: I would suggest to rephrase this as the 'recent decades' are already the leading words of the paragraph,  so 'at the same time' already incorporates it.
**RC2.03)** This is a good point, the section has been rephrased (line 21-24):
"The same period has experienced the emergence of Structural Health Monitoring (SHM), where data from sensors installed in a given structure is applied to inform Operation and Maintenance (O&M) strategies, in asset integrity assessments, and lately also for the assessment of potential life-extension through monitoring of strain histories at fatigue critical locations."

**RC2.04) (line 34)** From the abstract I remember the tower top is the location where the differences between MDE and aero-elastic results are biggest. Then the first statement in the introduction is about locations which are very far away from the top. This might be understood as if the problem you are looking at is not relevant?
**RC2.04)** This is an understandable comment, and we thank you for raising this concern. In direct relation to the comment, we have added an argument to why the global accuracy is relevant (line 30-32):
"Additionally, virtual sensing has the significant benefit of estimating the response of the structure at any location, hence not limiting the information from the Structural Health Monitoring System (SHMS) to a few predefined sensor locations.
Furthermore, we have added to the abstract a sentence about the error around the mean sea level, to highlight that an improved model might also improve the results here (line 7-9):
"Multi-band MDE estimates of section moments are compared to true responses in terms of Damage Equivalent Loads and Stresses. Long-term estimates show reduced accuracy in the area around the tower top and at ±15 m around the Mean Sea Level.

**RC2.05) (line 35)** I was a bit in doubt reading this absolute statement of inaccessibility of the inside of the monopile post/erection. I have then asked someone who is dealing with OWTs more than I do and they confirmed there is a hatch which should then allow to access the monopile. So perhaps it makes sense to rephrase this along the lines of 'only accessible with significant efforts'?
**RC2.05)** Thank you for this comment. We agree that it is possible to access the sensors in the monopile above the mud line. However, we believe that significant efforts would be made to avoid it. In a scenario where, for example, a fibre Bragg strain sensor needs to be replaced, it must be glued or welded on the inside of a submerged monopile with great accuracy. We are not entirely certain whether that is, in fact, possible or at least affordable.
The section has been modified to reflect the comment (line 24-26):
"However, for offshore structures, these critical locations are often sub-sea, where the strain sensors are only accessible with significant efforts, or sub-soil, where strain sensors cannot be installed or maintained in practice, post-erection."

**RC2.6) (line 36)** What about fibre brag sensors for strain monitoring? I agree the strain gauges have a drift and get damaged over the years, though.
**RC2.6)** Fibre Bragg sensors would, as you mention, not have the problem with drift or damage over time. However, for monitoring of strains/stresses in the monopile, the sensor would have to

be installed before the erection, thus making it vulnerable to damage during the installation. As the sensors are located sub-soil or sub-sea for this purpose, they are at least complicated and expensive to replace, and at worst, it is not possible.

**RC2.7) (line 38)** I wonder why this paper does not appear here: 10.1088/1742-6596/3025/1/012011 - isn't that very related?

**RC2.7)** Thank you for guiding our attention towards this paper. It does deal with virtual sensing, but the approach is data-driven, and it requires training data from strain gauges at the locations of the predicted response. This is very practical in case a critical sensor fails, however, it does not serve as a global monitoring strategy, which makes it less relevant for the case considered in the present paper. That being said, it is a good example of how machine learning can be used in virtual sensing and does add value in terms of background knowledge. Therefore, it has been added in the literature review in the introduction, in the section dealing with machine learning (line 38-41):

 "Lately, the use of neural networks has also entered the field of virtual sensing, e.g. when physics-guided learning from SCADA data and 10-minute acceleration statistics are used to estimate damage equivalent moments (de N Santos et al., 2023), or when virtual sensors are trained based on strain sensors for gap-filling in strain histories in case of sensor failure (Faria et al.,2025)."

**RC2.8) (line 91)** Besides the tiny remarks in the first paragraph, I find the rest of the introduction to be a very pleasant and easy-to-understand read. Good work!

**RC2.8)** Thank you for the nice comment.

**RC2.9) (line 173)** Again, a well written and understandable section without any comments.

**RC2.9)** Thank you for the nice comment.

**RC2.10) (Figure 2)** The generator torque seems to have the maximum values in blue and the minimum values in red, or am I mistaken?

**RC2.11) (Figure 2)** Just for the sake of beauty and clarity of the figures you might consider putting in rated wind speed as a vertical (dashed?) line. This is just optional and more a question of personal taste.

**RC2.10+RC2.11)** Thank you for bringing this to our attention. Figure 2 has been corrected accordingly, and so has Figure 3.

**RC2.12) (Figure 4)** Again, and similar to above, it might be nice to have the eigenmodes present as vertical lines in the plots.

**RC2.12)** Thank you for bringing this to our attention. We have included the first natural frequency of the first eigenmode in the figure 4(b), as this is the focus in the discussion of the figure.

**RC2.13) (Figure 5)** That is a beautiful figure!

**RC2.14)** Thank you for the nice comment.

**RC2.14) (line 342)** No comments to this section

**RC2.14)** Thank you for the nice comment.

**RC2.15) (Figure 8 (now 7))** Would it make sense to include the height of the TP as a horizontal line here?

**RC2.15)** Thank you for this comment. We have not included a horizontal line at the TP, but we have included the MSL and mud line as in similar figures in the paper.

**RC2.16) (Figure 8 (now 7))** This poor x-axis would love a label.
**RC2.16)** Thank you for bringing this to our attention. The figure has been corrected accordingly.

**RC2.17) (line 432)** A tiny remark on language here: The use of 'while' seems a bit out of place as it insinuates a timely connection or insinuates a contradiction and perhaps a split in two sentences as "... Ritz vectors. Similar methods..." is a simpler and more elegant solution.
**RC2.17)** Thank you for bringing this to our attention. The suggested correction has been made (line 374-375):
"Similar methods are applied in Iliopoulos et al. (2017), Augustyn et al. (2021), and Toftekær et al. (2023)."

**RC2.18) (line 450)** Is the thrust wind load on the tower negligible?
**RC2.18)** Thank you for this comment. The thrust wind load on the tower is likely not negligible, but its importance has not been assessed in detail. Instead (as described in section 5.1 (previously 4.1) ), the $1^{st}$ tower bending modes are included to represent the distributed wind load on the tower. We acknowledge the confusion of not mentioning this where the Ritz vectors are presented and have added a sentence in Section 4.3.2 (previously 3.3.2) referring to section 5.1 (previously 4.1) (line 392-394):
"A Ritz vector for the distributed wind load on the tower has not been established in the present work. However, as presented in Table 6, the first tower bending mode shapes are used to represent the quasi-static response resulting from this load."

**RC2.19) (Figure 9 (now 8))** This looks a bit like a linear, tilted line from bottom to sea level, is this supposed to look like it or should it look more like the non-linear shapes of the first two load cases.
**RC2.19)** Thank you for this comment. With the loading from the waves, which only acts on the submerged part of the monopile, the curvature of the beam elements is rather small. The deflected shapes in Figures 8 and 9 (previously 9 and 10) are in fact a correct representation of the deflected shape caused by the wave load.

**RC2.20) (Figure 10 (now 9))** Same as above, horizontal lines indicating important levels might add to the beauty of the Figure.
**RC2.20)** Thank you for this comment. We have included the MSL and mud line as in similar figures in the paper.

**RC2.21) (Figure 10 (now 9))** Albeit a normalized axis, this poor fella would really love to have a label as well.
**RC2.21)** Thank you for bringing this to our attention. The figure has been corrected accordingly.

**RC2.22) (Figure 11 now (10))** This is the point where I have to admit I am truly no expert in the area. It escapes me how we can get a large error of the MDE at the tower top when the part of the matrices that is measured is at the tower top? Maybe I am just very clueless and/or overlooked the explanation on this, but if there *is* a sensor this goes into the n_m array of measured values?
**RC2.22)** This is a valid comment, thank you. There is a sensor at the tower top, and it does go into the n_m array (in terms of a measured rotation for $f$ <0.05 Hz and a measured displacement for $f$>0.05 Hz). However, we estimate section moments based on nodal displacements and nodal rotations at both end nodes of the 3d beam element. This means that the measured DOFs only represent 2 out of the 8 (2 axial and 2 torsional, not important) DOFs that are used to calculate the section moment in the beam element. The structural response in the remaining 10

DOFs is obtained by modal expansion. At the tower top, the main contribution to the error comes from the frequency range $f>0.05$, hence, the large error arises because the relation between the measured displacement and the curvature of the mode shapes (particularly the 2nd and 3rd tower bending modes) does not match that of the IEA 15-MW RWT. We hope this reply clarifies why an error can occur at a measured location.

**RC2.23) (Figure 14)** As per the introduction of the paper, the levels below MSL might be the ones with difficult accessibility and thus the ones where MDE might be more useful. However, in this figure, the large error at TT cloud the understanding of the errors at lower levels. Would it make sense to split the figure to allow two x-axes?

**RC2.23)** Thank you for bringing this to our attention. It is indeed difficult to assess the errors at lower levels, and a good idea to split the figure. The figure has been corrected according to this comment as:

[Figure]

**Figure 14.** Error $\varepsilon_{MDE}$ of DESs for the MDE predicted section moment load histories in the FA (top) and SS (bottom) direction of the IEA 15-MW RWT from the individual HAWC2 simulation $s$, as presented in (32). Color gradient represents the mean wind speed at the hub $V_{hub}$ for the considered simulation $s$. Two separate x-axes are used to present $\varepsilon_{MDE}$ (illustrated with white and grey background colour).

**RC2.24) (line 650)**
**RC2.24)** thank you for the elaborate comments. We have replied to the individual comments below the respective sections:

This work is a well-written and understandable read with precise Figures. The reviewer congratulates the authors for its high level.
Thank you for this nice comment.

The only major point of critics concerns the motivation and relevance of the overall concept. The authors argue MDE is applied at positions not accessible for sensor application and than later show the significant errors to be close to the tower top. This position, however, seems rather easy to access and equip with sensors (as also mentioned by the authors and displayed in Figure 11). This contradiction seems unresolved to the reviewer and I ask the authors to comment on this.

Thank you for commenting on this. This is a valid concern, and the authors have attempted to clarify below:

- The multi-band MDE is a global monitoring approach, and thus it should perform accurately structure-wide, ideally, removing the need for location-specific sensors. In the reply to comment RC2.04, the global monitoring approach has been emphasised, and the error around the MWL is added to the argumentation for the need to include blade flexibility. Furthermore, the discussion has been extended with PSD plots of the MDE error around the MWL, which show that the error is related to erroneous tower mode shapes.
- Poor performance anywhere in the structure, broadly speaking, means that the mode shapes used in the MDE do not reflect the mode shapes of the actual structure. This can, in theory, lead to errors anywhere in the structure depending on the measured degrees of freedom.
- In a real-world scenario, it is often relevant to calibrate an FE model to measurement data. If the FE model disregards important parameters (such as blade flexibility), tuning parameters such as the soil stiffness are likely misinterpreted. The error in the natural frequency of the tower modes resulting from disregarding blade flexibility is discussed in (Reinhardt et al., 2024) which has also been included in the present paper (see RC1.06).

The authors hope that this reply satisfies the review comment and solves the contradiction.

Another aspect which might merit some additional explanation in writing: **1)** According to my understanding, DLC4.1 as per IEC61400 does not include turbulence. **2)** It's frequency in the standard also is significantly lower than its practical occurrence, mostly due to curtailment of power output as a function of energy trading. Recent presentations on the WESC 25 mentioned up to 50 turbine stops per day and also claimed this to be critical as they happen also under unfavorable operating conditions. I fully understand this cannot be covered with the present work, but I think a word or two in the conclusions to mention the possibility of DLC4.1 playing a much more significant role in real-world turbines would add value to it.

**1)** It is correct that DLC 4.1 does not include turbulence. However, in the simulations performed for the dataset (Pedersen et al., 2025), it was chosen to include turbulence and stochastic waves to obtain a more realistic time series response for this DLC (as well as for DLC 4.1). This is also explained in the documentation in (Pedersen et al., 2025).

**2)** A comment has been added in Section 3.2 (previously 2.5) regarding the duration of DLC 4.1 (line 222-224):

"However, in a real operating scenario, shut-down and start-up may have a larger influence on the lifetime damage, as they occur more frequently than described by IEC (2019b) due to, for example, curtailment. This has not been accounted for in the present paper."

and in the conclusion (line 598-600):

"The damage associated with start-up and particularly shut-down in normal conditions (DLCs 3.1 and 4.1) might be significantly underestimated in the present paper, as the durations specified by IEC (2019b) for these DLCs do not necessarily reflect a real operation scenario, where start-up and shut-down can occur for many reasons, including curtailment."

One last thing that escaped my understanding and where I merely ask for a layman's explanation is the seemingly contradiction of the tower top sensor being there and the huge error of MDE at tower top (also pointed to in a dedicated comment).

We have replied to this under comment **RC2.22.**

Ghoshal, A. (2024). *Colossal 20-MW wind turbine is the largest on the planet (for now)*. https://newatlas.com/energy/world-largest-offshore-wind-turbine-20-mw-mingyang/

Pedersen, M. G., Rinker, J., Høgsberg, J., & Farreras, I. A. (2025). *IEA-15MW-RWT-Monopile HAWC2 Response Database*. Technical University of Denmark. https://doi.org/10.11583/DTU.24460090.v3

Reinhardt, T., Sastre Jurado, C., Weijtjens, W., & Devriendt, C. (2024). On the influence of rotor nacelle assembly modelling on the computed eigenfrequencies of offshore wind turbines. *Journal of Physics: Conference Series*, *2767*(5). https://doi.org/10.1088/1742-6596/2767/5/052034

Salas, J. (2025). *Another turbine world record set – but not by China this time*. https://newatlas.com/energy/siemens-gamesa-sg-dd-276-turbine/